# Hypothesis-Driven Feature Manifold Analysis in LLMs via Supervised Multi-Dimensional Scaling

**Federico Tiblias**[1,2,3,4]      *federico.tiblias@tu-darmstadt.de*

**Irina Bigoulaeva**[1,2,3]      *irina.bigoulaeva@tu-darmstadt.de*

**Jingcheng Niu**[1,2,3]      *jingcheng.niu@tu-darmstadt.de*

**Simone Balloccu**[1]      *simone.balloccu@tu-darmstadt.de*

**Iryna Gurevych**[1,2,3]      *iryna.gurevych@tu-darmstadt.de*

[1] *Department of Computer Science, Technical University of Darmstadt*
[2] *Ubiquitous Knowledge Processing Lab (UKP Lab)*
[3] *National Research Center for Applied Cybersecurity ATHENE, Germany*
[4] *Zuse School ELIZA, Technical University of Darmstadt*

**Reviewed on OpenReview:** https://openreview.net/forum?id=vCKZ40YYPr

## Abstract

The linear representation hypothesis states that language models (LMs) encode concepts as directions in their latent space, forming organized, multidimensional manifolds. Prior work has largely focused on identifying specific geometries for individual features, limiting its ability to generalize. We introduce Supervised Multi-Dimensional Scaling (SMDS), a model-agnostic method for evaluating and comparing competing feature manifold hypotheses. We apply SMDS to temporal reasoning as a case study and find that different features instantiate distinct geometric structures, including circles, lines, and clusters. SMDS reveals several consistent characteristics of these structures: they reflect the semantic properties of the concepts they represent, remain stable across model families and sizes, actively support reasoning, and dynamically reshape in response to contextual changes. Together, our findings shed light on the functional role of feature manifolds, supporting a model of entity-based reasoning in which LMs encode and transform structured representations. Our code is publicly available at: https://github.com/UKPLab/tmlr2026-manifold-analysis.

## 1 Introduction

There is increasing evidence from recent work in mechanistic interpretability that language models develop structured representations of entities in their latent space. Notably, Heinzerling & Inui (2024) find that numerical entities (e.g., `Karl Popper was born in 1902`) are represented in a monotonic, "pseudo-linear" fashion. Increasing or decreasing specific neuron activations can lead the model to output a higher or lower value. More recently, Engels et al. (2025) discover non-linear modes of structural entity representation, which form strikingly interpretable patterns. They show that days of the week (`Sunday`, `Monday`) and months (`December`, `January`), for example, form a circular structure. Concurrent work by Modell et al. (2025) provides formal definitions of these *feature manifolds* and explores how they arise in LMs.

Nevertheless, several fundamental questions remain unanswered: we do not know if and how LMs make use of these manifolds during reasoning, or how to reliably detect their presence (Engels et al., 2025; Modell et al., 2025). Answering these questions can help improve LMs and how we control them. This is particularly important in light of current limitations of LMs, such as poor temporal reasoning (Yuan et al., 2023; Niu

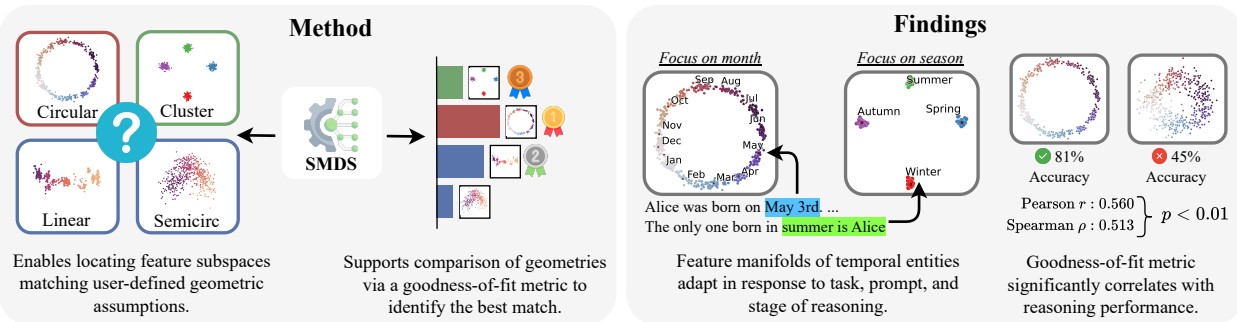

Figure 1: **Our Main Contributions**. Supervised Multi-Dimensional Scaling is a novel dimensionality reduction method for identifying and testing subspaces with known geometry (left). Using it, we show that temporal entities in LMs form task- and prompt-dependent *feature manifolds* that support reasoning (right).

et al., 2024), difficulty in alignment (Wang et al., 2023), bias (Gallegos et al., 2024), and vulnerability to distraction (Shi et al., 2023; Niu et al., 2025).

In this paper, we address these questions by introducing **Supervised Multi-Dimensional Scaling** (**SMDS**), a novel method to systematically analyze feature manifolds. Commonly used dimensionality reduction methods typically enforce fixed structural assumptions, making it difficult to compare results across alternative geometric hypotheses. SMDS complements existing methods by providing a unified way to specify arbitrary geometric assumptions and a quantitative metric to evaluate their fit. SMDS reframes the identification of a feature manifold as a model selection problem, and thus offers quantitative support for claims about the underlying structure of learned representations. Moreover, this method enables observing how a feature manifold evolves across different layers and reasoning steps.

We focus on temporal reasoning through short-form QA tasks, such as identifying recency, ordering events and estimating durations, as we consider them an ideal test bed for manifold analysis. This choice is motivated by three factors: (1) LMs display poor performance in such tasks (Yuan et al., 2023; Huang et al., 2023; Niu et al., 2024); (2) initial evidence has found temporal feature manifolds to vary widely across tasks (Heinzerling & Inui, 2024; Engels et al., 2025); and finally, (3) there is a gap in analyses targeting the *atomic structures* of temporal reasoning from a mechanistic standpoint.

Our main findings can be summarized as follows:

$\mathcal{F}_1$: **Temporal entities form feature manifolds with intuitive structures, and this pattern is consistent across model architectures and sizes.** We find that the manifolds associated with various temporal concepts (e.g., days of the year, hours, durations, and historical events) align with interpretable geometries such as circles, lines, and clusters, thus substantially extending Engels et al.'s (2025) findings. Our SMDS experiments cover over sixty thousand recovered manifolds and confirm that the identified feature structures are robustly shared across different model sizes and architectures.

$\mathcal{F}_2$: **Feature manifolds are dynamically adjusted depending on the task.** SMDS enables us to compare manifold structures across different token positions. We analyze prompts that share the same context but differ in their final completion cue, and find that LMs alter feature manifolds based on the cue and task in an intuitive way.

$\mathcal{F}_3$: **Feature manifolds actively support reasoning.** We find that LMs actively utilize feature manifolds to perform reasoning tasks, supported by two pieces of crucial evidence. First, perturbing manifold-aligned subspaces consistently impairs reasoning performance, while equivalent noise applied to random subspaces has a negligible effect. Second, we observe that manifold quality significantly correlates with downstream performance.

When combined with previous results on the binding problem (Feng & Steinhardt, 2023), our findings suggest an explanation for the mechanism by which LMs perform reasoning. We hypothesize an **entity-based reasoning pipeline** in LMs that:

1. Represents entity properties in coherent locations on a manifold within the residual stream;

2. Applies a transformation to this manifold, guided by the question or task context;

3. Selects an appropriate output based on the transformed representation.

Finally, we extend our analysis beyond mono-dimensional temporal features into two separate experiments (§5.4): the first is an entity-based reasoning task on geography that similarly uncovers manifold structures shared across models; the second studies a pair of temporal features to locate a multidimensional manifold. These experiments show our analysis can be extended beyond the temporal domain and to higher-dimensional features. Overall, these results suggest that feature manifolds play an important role in how LMs represent and reason about entities. We view this work as a step towards better understanding the mechanisms behind reasoning in modern language models.

**Contributions**  We first present a survey of previous feature manifold analysis methods and the types of geometric structure they are able to capture (§2). We then introduce the novel SMDS method in §3. Next, we present our results in §5, where we outline three major findings: (§5.1) manifold geometry for the same type of entity is shared across models; (§5.2) LMs adapt structures in context for different tasks; and (§5.3) LMs actively use feature manifolds for reasoning. Moreover, we show that our approach extends to other domains and to multidimensional manifolds (§5.4). Finally, we conclude the paper with a discussion (§6).

## 2 Feature Manifold Analysis

Dimensionality reduction methods commonly used in manifold analysis typically rely on fixed structural assumptions about the data, which makes it difficult to quantitatively compare alternative geometric hypotheses. This gap motivates us to introduce our SMDS method in Section 3. In this section, we set up the problem with relevant background and survey existing feature manifold analysis methods.

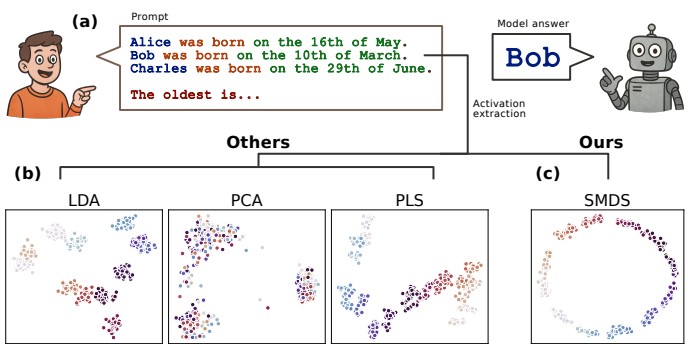

Figure 2: Comparison of structural assumptions in LDA, PCA, PLS and SMDS with a circular geometry specified as the hypothesis.

**Preliminaries**  We illustrate our method using a temporal reasoning task as a running example (Figure 2a). Performing temporal reasoning requires a model to understand both explicit mentions of temporal expressions (Jia et al., 2018b) and implicit knowledge of temporal calculus (Allen, 1981). Our analysis focuses on how LMs process temporal expressions, which are central to temporal reasoning and define precise, measurable quantities that can reveal underlying feature manifolds. Temporal reasoning also offers good diversity: different types of temporal expressions demand different reasoning skills (e.g., comparing frequencies, ordering events, or identifying recency) and models vary widely in how well they handle these tasks (Chu et al., 2024).

In particular, we seek to start from confirming Engels et al.'s (2025) finding that LMs tend to represent calendar dates in a circular topology, placing December near January in their latent space. Consider a prompt comprising several sentences following the template "**<name> was born on the <day> of <month>.**" When asked "**The oldest is,**" the task is answered correctly if the model uses contextual information to produce the correct answer **<name>**.

By querying the LM with several such prompts varying the reference date, we elicit internal representations that collectively reside on the feature manifold of calendar dates. In this case, our quantity of interest is

the birthday of the correct person (e.g., Bob's birthday: `10th of March` in Figure 2a), which we collectively represent as a set of labels $y$. We map these labels onto the $[0, 1]$ interval, where 0 corresponds to Jan 1st and 1 to Dec 31st. We then extract the hidden states corresponding to the last token of the date (e.g., the "`<month>`" token[1]), yielding a collection of hidden states $X \in \mathbb{R}^{n \times d}$, with $n$ number of samples and $d$ the hidden size of the LM. Next, we use dimensionality reduction to project the high-dimensional hidden states $X$ onto an interpretable, low-dimensional space.

**Existing Methods**  We identify three primary methods used in previous works: PCA, LDA, and PLS (Wold et al., 2001; Park et al., 2024a; El-Shangiti et al., 2025; Modell et al., 2025, *inter alia*).[2] From observing the visualizations in Figure 2b, we see that each method highlights different aspects of the underlying data structure but none can readily recover arbitrary geometries such as the circular one we seek. LDA finds interpretable clusters but has no notion of order; PCA fails to identify feature subspaces if they are not aligned with the directions of maximum variance; and PLS is limited to linear features unless a suitable transformation is applied to the data (AlquBoj et al., 2025). Moreover, without quantitative metrics to assess the goodness-of-fit across different methods, it is unclear which of the manifolds best reflects the original representations.

## 3 Supervised Multi-Dimensional Scaling

To address this, we introduce Supervised Multi-Dimensional Scaling (SMDS), a method for testing and comparing user-specified geometric hypotheses about representation structure. It extends classical Multi-Dimensional Scaling (MDS; Ghojogh et al., 2020) by incorporating supervision, under the assumption that labels can parametrize the underlying feature manifold formed by the model's hidden states. The method first uses MDS to build an ideal geometry representing the manifold, and then learns a linear mapping from model embedding to this manifold structure. SMDS is flexible, as varying the assumption enables recovering multiple different structures, and provides a common basis to quantify their fit and thus identify a preferential one.

Formally, we assume that activations $X$ forming the feature manifold can be located using labels $y$ that represent a numerical property. SMDS first computes ideal pairwise distances $d(y_i, y_j)$ between $y_i, y_j \in y$ that encode the geometry of the desired manifold (e.g., circular, linear, or clustered). It then finds a linear projection $W \in \mathbb{R}^{m \times d}$ such that the Euclidean distances between projected points $W x_i$ and $W x_j$ best match $d(y_i, y_j)$, with $x_i, x_j \in X$. SMDS minimises the loss:

$$\mathcal{L} = \sum_{i<j} \left( \|W(x_i - x_j)\|^2 - d(y_i, y_j)^2 \right)^2. \tag{1}$$

$d(y_i, y_j)$ is task-dependent and implicitly defines the hypothesis structure. For example,

$$d(y_i, y_j) := 2 \sin \left( \pi \min \left( \delta_{ij}, \ 1 - \delta_{ij} \right) \right), \quad \delta_{ij} := |y_i - y_j|, \tag{2}$$

these two formulas represent the chord distance between two points on a unit circle, thereby defining a *circular* structure. As shown in Figure 2c, SMDS finds a clear circular projection of calendar dates, consistent with Engels et al.'s (2025) findings.

We assess the quality of a recovered projection $W$ trained on activations $X$ by computing a variant of normalized stress (Amorim et al., 2014), adapted for a supervised task. In particular, we compute stress over a held-out set of points $\hat{X}, \hat{y}$ and corresponding ideal distances $\hat{d}_{ij} = d(\hat{y_i}, \hat{y_j})$:

$$S := \sum_{i<j} \left[ \|W\hat{x}_i - W\hat{x}_j\| - \hat{d}_{ij} \right]^2 / \sum_{i<j} \hat{d}_{ij}^2. \tag{3}$$

This metric measures how well distances in the recovered projection align with those of a hypothesized manifold. High-dimensional activations that originally exhibit a particular geometric structure can often

---

[1] For readability, we omit space tokens in the examples. Tokenization is still performed as usual.
[2] We provide a review of relevant works in §A.

be projected into a low-dimensional space that preserves this geometry, thereby yielding low stress. By comparing stress values across multiple distance functions, one can identify the best-fitting manifold. In practice, however, selecting a single best hypothesis can be challenging, as stress scores for different manifolds frequently cluster closely together. In such cases, additional evidence is needed to discriminate among candidates, for example via statistical testing, as performed in §5.

**Distance Functions**   We propose a set of distance functions for SMDS to detect a heterogeneous variety of manifolds. Seminal works have shown several instances of the idiosyncratic structure of feature manifolds. Notable examples include:

- Cyclical features form a ring shape in the latent space (Engels et al., 2025);
- Numbers are compressed according to a logarithmic progression (AlquBoj et al., 2025);
- Years of the 20th century form a U-shaped structure (Engels et al., 2025; Modell et al., 2025);
- Categorical features visually form clusters corresponding to the vertices of a polytope (Park et al., 2024a);
- Lastly, Gurnee & Tegmark (2023) have extracted multidimensional manifolds representing features such as latitude and longitude.

Therefore, as listed in Table 1, we parametrize shapes such as circles, semicircles, lines, logarithmic lines and clusters so that the resulting manifold is interpretable. The manifolds we define are categorized based on their topology: linear, where concepts follow a continuous, monotonic progression; cyclical, where the progression is continuous but wraps around to the starting point, forming a loop; and categorical, where concepts occupy discrete, equidistant regions without inherent ordering.

In the following sections, we use this collection of distance functions to identify feature manifolds for several tasks and at different stages of the reasoning process.

Table 1: Collection of distance functions used throughout our study. colors denote manifold topology: `linear`, `cyclical` or `categorical`. $\delta_{ij} \coloneqq y_i - y_j$. $M \coloneqq \max(y)$.

| Distance Function $d(y_i, y_j)$ | Resulting Manifold |
|---|---|
| $\|\delta_{ij}\|$ | `linear` |
| $\|\log y_i - \log y_j\|$ | `log_linear` |
| $2\sin(\frac{\pi}{2}\|\delta_{ij}\|)$ | `semicircular` |
| $2\sin(\frac{\pi}{2}\|\log y_i - \log y_j\|)$ | `log_semicircular` |
| $2\sin(\pi \min(\|\delta_{ij}\|, 1 - \|\delta_{ij}\|))$ | `circular` |
| $\min(\|\delta_{ij}\|, M + 1 - \|\delta_{ij}\|)$ | `discrete_circular` |
| $0$ if $y_i = y_j$, $1$ otherwise | `cluster` |

## 4   Experimental Setup

**Data & Prompt Setup**   Based on the TIMEX3 specification (Pustejovsky et al., 2010), we create five synthetic datasets and three variants, probing precise aspects of temporal understanding over a variety of numerical quantities (Table 2). All sentences across datasets have a similar format: they describe an action performed by three individuals, the action is associated with a temporal expression, and a continuation cue is attached to elicit temporal reasoning. The right answer is always one of the three names mentioned in the context. We randomize the names, actions, and temporal expressions to increase robustness but keep the same structure across all samples. Temporal expressions are sampled uniformly across a given range, but respecting some plausibility constraints (e.g. "`once per year`" is never associated with common actions such as "`takes a shower`"). We also make sure names are always tokenized as a single token for all models. The three variants date_season, date_temperature, and time_of_day_phase share the same context and range with their main counterparts, but ask a different question that requires a different type of reasoning (e.g. "`The only person born in spring is`"). Overall, our data exhibits greater variability than similar datasets used in previous literature. See Appendix B for an extended discussion on our temporal taxonomy, datasets and for the variability analysis.

**LM Selection**   The bulk of our analysis is performed on three models from different families: Qwen2.5-3B-Instruct (Team et al., 2025), Llama-3.2-3B-Instruct (Grattafiori et al., 2024), gemma-2-2b-it (Gemma et al., 2025). We also study what impact instruction tuning has on these representations by comparing these models with their base versions. For the Llama family, we also study larger models to observe whether the manifolds

Table 2: Tasks and Corresponding Prompts. Variants `date_season`, `date_temperature`, and `time_of_day_phase` are omitted for brevity and are detailed in Appendix B. colors represent templates: **blue** denotes names, **orange** denotes actions, **red** denotes the corresponding continuations, **green** denotes temporal expressions, and **black** denotes expressions that do not change throughout the dataset.

| Dataset | Context | Continuation | Expression Range |
|---------|---------|--------------|------------------|
| date | Anna took a bus on the 16th of January. | The first person that took a bus was | 01/01 – 31/12 |
| duration | Neil is starting a workshop on the 11th of January lasting 1 day. | The person whose workshop ends first is | 01/01 – 31/12
1 day – 4 years |
| notable | Emma was born on the day Pius X became Pope. | The oldest is | 1900 – 2000 |
| periodic | Kevin waters the plants every day. | The person who waters the plants more often is | daily – every 6 years |
| time_of_day | Lucy naps at 16:15. | It is now 19:37. The last person who napped is | 00:00 – 23:59 |

we identify persist at scale: Llama-3.1-8B-Instruct and Llama-3.1-70B-Instruct. Due to computational constraints, we run these models using 4-bit quantization.

**Generalizing Manifold Analysis**  We generalize our study by analyzing activations across all layers and at different positions along the sentence. In particular, we consider three sites: (i) the final token of the temporal expression (e.g., "**on the 16th of January**," abbreviated as TE); (ii) the final token of the prompt (e.g., "**The first person that took a bus was**," abbreviated as LP); and (iii) the token corresponding to the generated answer (i.e., one of the contextual names, abbreviated as A).

To systematize the hypothesis selection process, we drop any assumption about which manifold should correspond to which feature and instead run a grid search over all defined distance functions. We fit an instance of SMDS for each dataset and layer and then compare recovered manifolds using stress (Eq. 3). Throughout the study, we choose $m = 3$ as it is the minimum number of dimensions required to represent all our hypothesis manifolds (1D for most linear manifolds, 2D for some linear and cyclical ones, and 3D for clusters, which form a tetrahedron in 3D space). Increasing dimensionality yields similar results, which are discussed in Appendix D.4. Unless stated otherwise, all manifolds visualized in the study show the first two components identified by SMDS for the best-scoring layer, computed with a 50/50 train/test split.

To demonstrate the robustness of our analysis, we conduct a statistical significance study using a 10-fold cross-validation repeated 5 times across all datasets, models, and manifolds. First, we group observations by dataset, model, and manifold rank, and perform a Friedman test. Then, for all groups that achieve statistical significance ($p < 0.05$), we perform a post-hoc Nemenyi test to evaluate the significance of manifold ranks on a given dataset. To break any ties in manifold rank, we additionally perform bootstrapping with 500 iterations, followed by the same Friedman–Nemenyi protocol as before. This yields even stronger statistical guarantees for the identified hypotheses. We define a preferential manifold as one that (1) attains the best average score and (2) is statistically significantly different from all alternative hypotheses. Further details on statistical significance are provided in Appendix D.3.

## 5   Experiment Results and Analysis

We present our experiment results around the three major findings in this section.

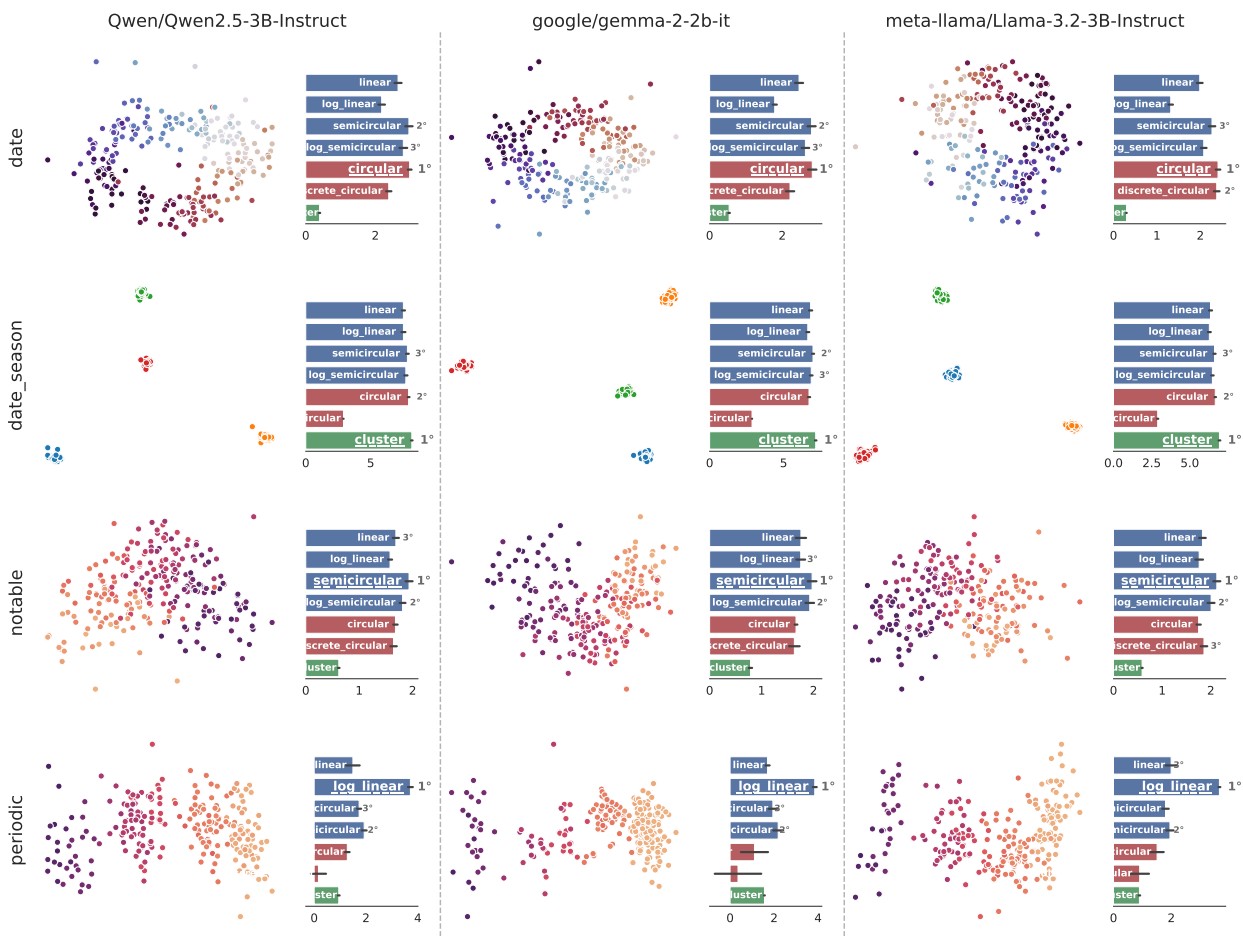

Figure 3: Feature manifolds retrieved from the LP site. We can observe that models represent features in a similar way, and the resulting manifolds are interpretable and match an intuitive progression (linear, circular or categorical) of the underlying features. The scatter plots on the **left** show the first two components of SMDS dimensionality reduction; the bar plots on the **right** depict scoring of different manifolds on the given activations. Scores displayed are computed as $-\log S$ to emphasize the difference between values; error bars are shown in black. Bar plot color reflects manifold topology: ■ linear; ■ cyclical; ■ categorical;

### 5.1 ($\mathcal{F}_1$) Temporal Entities Share Intuitive Manifold Structures Across Models.

Figure 3 and Table 3 show the best-scoring manifolds across models and tasks. We first observe that all manifolds identified this way are not only interpretable, but also match prior research (Engels et al., 2025; Park et al., 2024a; AlquBoj et al., 2025). Their topology always matches meaningful properties of the feature they explain: monotonic features are represented by linear topologies, cyclical features wrap around in loops, and categorical features map to cluster structures. Notably, the best manifold shape is consistent across all observed model families as well as in most of the non-instruction-tuned counterparts. Moreover, this pattern persists at scale, with all three observed sizes (3B, 8B, 70B) creating coherent shapes between them. This suggests there are preferential ways to encode the same knowledge, and all language models eventually converge to similar structures, providing further proof of hypotheses formulated in previous literature (Huh et al., 2024).

Previous work has shown that LMs encode numerical quantities in a logarithmically compressed way (AlquBoj et al., 2025). Our work extends this finding to temporal reasoning for the first time: in both the duration and periodic tasks, time intervals such as days to weeks, weeks to months, and months to years are preferentially represented with roughly uniform spacing, indicating a logarithmic compression of temporal magnitude.

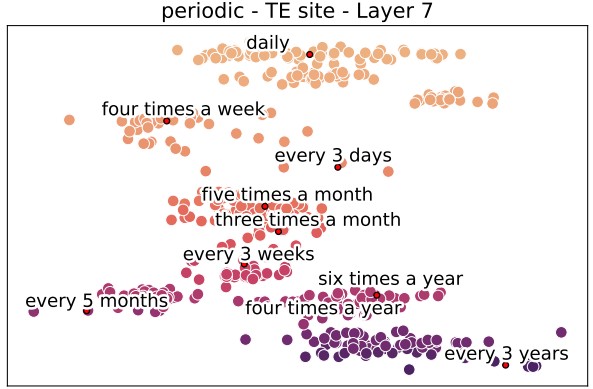

Figure 4: Llama-3.2-3B-Instruct on the `periodic` task. Events display logarithmic compression in their frequency: long intervals (e.g., months, years) are represented with the same granularity as shorter ones (e.g., days, weeks).

Critical difference diagrams across tasks

Figure 5: Avg. rank of manifolds across all models over 500 bootstrapping iterations. Horizontal bars show groups of statistical equivalence. Best-ranking manifold is always statistically different from others.

This pattern (Figure 4), though not directly comparable, bears a superficial resemblance to the logarithmic compression described by the Weber–Fechner law (Dehaene, 2003). We note that the labels we use are themselves logarithmically spaced. A deeper study into temporal understanding is therefore needed to clarify whether the compression we observe is a genuine emergent behavior or an artifact of our synthetic dataset.

Many tasks exhibit high scores for more than one topology. The cross-validation setup is not sufficient to break all ties on manifold rank, therefore we perform bootstrapping with 500 iterations. Figures 5, and 17 show the overall ranking of manifolds across tasks confirming the results obtained in the previous analysis and establishing a clear best-ranking hypothesis for each task.

This analysis both validates SMDS as a manifold analysis tool and underscores the difficulty of identifying a single best-ranking manifold for a given problem. One plausible explanation is that, while preferential manifolds do exist, models often construct multiple valid representations whose disentanglement requires

Table 3: Best-scoring manifold, corresponding average stress, and accuracy for different models and tasks in a 10-fold setting with 5 repetitions. Standard error shown in grey.

| | date $-\log S$ | date Manifold | date Acc | date_season $-\log S$ | date_season Manifold | date_season Acc | date_temperature $-\log S$ | date_temperature Manifold | date_temperature Acc | duration $-\log S$ | duration Manifold | duration Acc |
|---|---|---|---|---|---|---|---|---|---|---|---|---|
| Llama-3.1-70B-IT | $2.909_{\pm0.017}$ | circ | 0.93 | $6.047_{\pm0.091}$ | clust | 0.83 | $6.219_{\pm0.105}$ | log_semic | 0.82 | $3.194_{\pm0.025}$ | log_lin | 0.62 |
| Llama-3.1-8B-IT | $2.483_{\pm0.020}$ | circ | 0.39 | $6.628_{\pm0.030}$ | clust | 0.66 | $6.018_{\pm0.038}$ | clust | 0.49 | $2.990_{\pm0.044}$ | log_lin | 0.09 |
| Llama-3.2-3B-IT | $2.502_{\pm0.022}$ | circ | 0.81 | $7.031_{\pm0.029}$ | clust | 0.74 | $6.525_{\pm0.040}$ | clust | 0.51 | $3.124_{\pm0.029}$ | log_lin | 0.30 |
| Qwen2.5-3B-IT | $3.267_{\pm0.031}$ | circ | 0.36 | $8.502_{\pm0.046}$ | clust | 0.26 | $7.548_{\pm0.049}$ | log_semic | 0.25 | $2.805_{\pm0.030}$ | log_lin | 0.32 |
| gemma-2-2b-IT | $2.969_{\pm0.028}$ | circ | 0.38 | $7.408_{\pm0.032}$ | clust | 0.29 | $7.207_{\pm0.058}$ | clust | 0.36 | $3.390_{\pm0.030}$ | log_lin | 0.26 |
| Llama-3.2-3B | $2.116_{\pm0.022}$ | disc_circ | 0.39 | $6.780_{\pm0.025}$ | clust | 0.53 | $6.702_{\pm0.040}$ | log_semic | 0.37 | $2.879_{\pm0.029}$ | log_lin | 0.21 |
| Qwen2.5-3B | $2.797_{\pm0.059}$ | circ | 0.21 | $8.439_{\pm0.041}$ | clust | 0.55 | $7.254_{\pm0.050}$ | log_lin | 0.23 | $2.959_{\pm0.028}$ | log_lin | 0.17 |
| gemma-2-2b | $3.142_{\pm0.030}$ | circ | 0.31 | $7.358_{\pm0.025}$ | clust | 0.61 | $6.948_{\pm0.050}$ | clust | 0.33 | $2.721_{\pm0.041}$ | log_lin | 0.11 |

| | notable $-\log S$ | notable Manifold | notable Acc | periodic $-\log S$ | periodic Manifold | periodic Acc | time_of_day $-\log S$ | time_of_day Manifold | time_of_day Acc | time_of_day_phase $-\log S$ | time_of_day_phase Manifold | time_of_day_phase Acc |
|---|---|---|---|---|---|---|---|---|---|---|---|---|
| Llama-3.1-70B-IT | $2.636_{\pm0.071}$ | semic | 0.18 | $3.858_{\pm0.052}$ | log_lin | 0.60 | $1.482_{\pm0.015}$ | circ | 0.53 | $6.182_{\pm0.070}$ | clust | 0.73 |
| Llama-3.1-8B-IT | $2.270_{\pm0.030}$ | semic | 0.31 | $3.882_{\pm0.036}$ | log_lin | 0.28 | $1.298_{\pm0.018}$ | circ | 0.12 | $6.731_{\pm0.026}$ | clust | 0.69 |
| Llama-3.2-3B-IT | $2.192_{\pm0.026}$ | semic | 0.49 | $3.734_{\pm0.032}$ | log_lin | 0.46 | $1.278_{\pm0.014}$ | circ | 0.30 | $7.072_{\pm0.029}$ | clust | 0.66 |
| Qwen2.5-3B-IT | $1.964_{\pm0.028}$ | semic | 0.32 | $3.804_{\pm0.032}$ | log_lin | 0.32 | $1.140_{\pm0.014}$ | circ | 0.19 | $8.803_{\pm0.030}$ | clust | 0.30 |
| gemma-2-2b-IT | $2.036_{\pm0.023}$ | semic | 0.58 | $3.929_{\pm0.041}$ | log_lin | 0.14 | $1.256_{\pm0.032}$ | semic | 0.07 | $7.589_{\pm0.035}$ | clust | 0.29 |
| Llama-3.2-3B | $1.103_{\pm0.181}$ | disc_circ | 0.01 | $3.600_{\pm0.033}$ | log_lin | 0.33 | $1.284_{\pm0.020}$ | circ | 0.10 | $7.061_{\pm0.028}$ | clust | 0.64 |
| Qwen2.5-3B | $1.764_{\pm0.048}$ | semic | 0.08 | $3.436_{\pm0.027}$ | log_lin | 0.18 | $1.287_{\pm0.012}$ | circ | 0.24 | $8.711_{\pm0.030}$ | clust | 0.45 |
| gemma-2-2b | $1.423_{\pm0.315}$ | log_lin | 0.01 | $3.769_{\pm0.039}$ | log_lin | 0.30 | $1.507_{\pm0.016}$ | circ | 0.18 | $7.599_{\pm0.030}$ | clust | 0.60 |

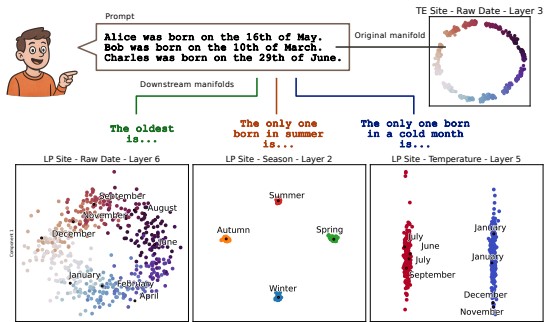
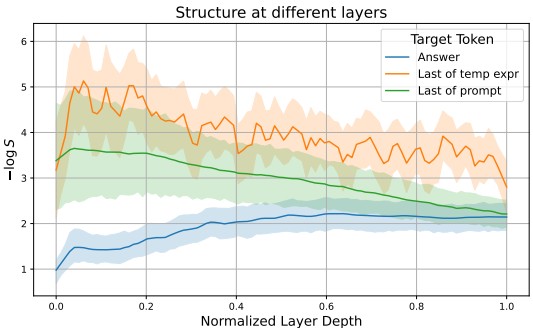

Figure 6: Feature manifolds of Llama-3.2-3B-Instruct on the `date` task and its variants. Best-scoring layers shown, as identified in Section 4. Different continuations produce drastically different topologies.

Figure 7: Manifold quality at different layers and positions in the sentence. Information transits from its injection point (orange) to the answer token (blue).

substantial computation. This representational polymorphism is not an artefact of SMDS: control tasks with randomized labels exhibit consistently high stress, confirming that SMDS does not simply overfit a hypothesized manifold. We provide a more detailed discussion in Appendix D.7.

### 5.2 ($\mathcal{F}_2$) LMs Adapt Structures In-Context for Different Tasks.

This section describes two observed phenomena in which LMs reshape manifolds across tasks and depth.

Figure 6 shows how the LM adapts the TE site feature manifold to different structures at the LP site, depending on the question prompt. Tasks `date`, `date_season` and `date_temperature` all start from the same context but result in strikingly different final structures: in `date`, a circular structure is required to account for the looping nature of dates in a year, while in the other two tasks inputs are mapped to linearly separable clusters. This can be interpreted as the model internally performing regression or classification to solve the task.

When comparing the location in the sentence where the structure is located, models exhibit a form of information flow between entities, which can strengthen certain manifold structures, degrade others, or even drastically change their shape as observed earlier. Figure 7 shows how to detect this flow with stress. In initial layers, the TA site is highly structured. As layers progress, this structure disperses into later tokens, such as the LP token and the A token. This process is not perfect: duplicated manifolds on LP and A display noticeably higher stress than the ones found at the TE site. A possibility is that later tokens in the same sentence accumulate more contextual information than early ones, thus resulting in noisier manifolds. Our results extend previous findings on the existence of a binding mechanism in LMs (Feng & Steinhardt, 2023; Dai et al., 2024): we show that not only vectors, but entire feature manifolds are preserved and propagated between entities.

### 5.3 ($\mathcal{F}_3$) LMs Actively Use Feature Manifolds for Reasoning.

Here we present two causally relevant lines of evidence that LMs actively use the structure of their representations to perform temporal reasoning.

**Located subspaces are causally relevant to noise perturbation.** To demonstrate that feature manifolds are utilized by LMs in their reasoning process, we perform causal intervention by adding noise to the manifold subspace and measuring downstream accuracy. We inject Gaussian noise $\epsilon \sim \mathcal{N}(0, \sigma^2 I_m)$ into the first layer at the TE site. Given a hidden state $x \in \mathbb{R}^d$, the perturbation is applied as $x' = x + W^{-1}\epsilon$, where $\epsilon$ is an $m$-dimensional noise vector projected back into the original space. Subspaces of dimension $m$ are located via SMDS in the usual way, and overfitting is prevented by training and evaluating the SMDS on a 50/50 split. We select the top three task-model pairs achieving the best accuracy on the original task, as

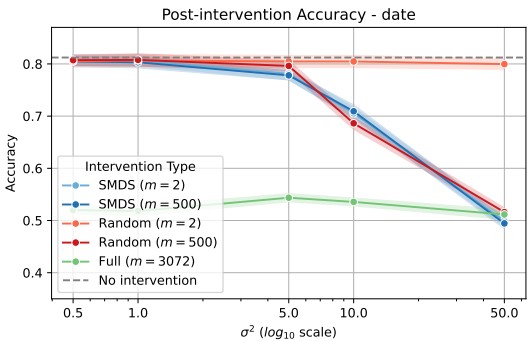

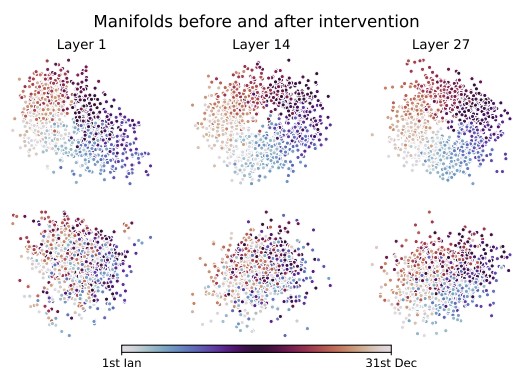

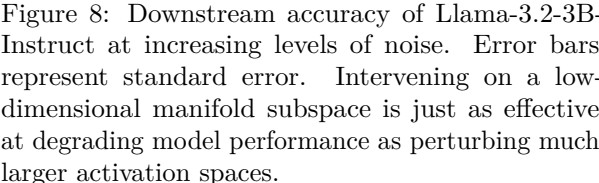

Figure 8: Downstream accuracy of Llama-3.2-3B-Instruct at increasing levels of noise. Error bars represent standard error. Intervening on a low-dimensional manifold subspace is just as effective at degrading model performance as perturbing much larger activation spaces.

Figure 9: Llama-3.2-3B-Instruct on the `date` task. Latent space of the LP token before and after applying noise on the TE token (top and bottom respectively). Interventions on early tokens cause disruptions to the manifolds of later ones.

these will be the settings where a disruption will be more noticeable: `date`, `date_season` and `time_of_day_phase` on Llama-3.2-3B-Instruct.

Across all tasks, performance gracefully degrades as the noise scale is increased (Figure 8). Crucially, we observe degradation for $m$ as low as 2, suggesting that temporal features are concentrated in very small yet highly informative regions of the activation space. We perform two other types of intervention in which we inject noise in the full latent space and in a random subspace, respectively. Affecting the full latent space achieves a much more destructive effect for low values of $\sigma^2$. On the other hand, disrupting a random subspace has no detectable effect on performance for subspaces of size $< 100$. The addition of noise also results in the disruption of structures located at subsequent tokens and layers (Figure 9). Interestingly, later layers are still able to form a vaguely organized shape, meaning information is partially being propagated or reconstructed. Our choice of perturbing the first layer is empirically motivated by the fact that intervention on later layers did not show as strong an effect. We hypothesize this is because information propagates quickly across tokens and layers, therefore the model is able to reconstruct a manifold from context tokens even if its source token has been disrupted. Overall, our experiments confirm that SMDS-located subspaces are critical for temporal understanding.

**Manifold quality significantly correlates with model performance.** We find a significant positive correlation between downstream accuracy and the ability of models to form well-organized manifolds, as quantified by $-\log S$ (Spearman's $\rho = 0.513$, $p = 0.0174$; Pearson's $r = 0.560$, $p = 0.0083$). Notably, this relationship emerges only for models that attain above-chance accuracy, specifically, Llama-3.2-3B-Instruct, Llama-3.1-8B-Instruct, and Llama-3.1-70B-Instruct. The results suggest that while feature manifolds tend to emerge naturally in LMs, a critical factor for strong performance lies in how effectively the model utilizes them during reasoning.

Table 4: Stress for the `duration` task at the LT site evaluated with the **linear** manifold. Standard error shown in grey. Differences with a control task with randomized labels are all statistically significant.

| Model | Best Layer | $-\log S$ | Control $-\log S$ | $p$ |
|---|---|---|---|---|
| Llama-3.2-3B-IT | 20 | $2.173_{\pm 0.114}$ | $0.469_{\pm 0.020}$ | 0.031 |
| Qwen2.5-3B-IT | 2 | $2.585_{\pm 0.047}$ | $0.506_{\pm 0.024}$ | 0.031 |
| gemma-2-2b-it | 2 | $2.595_{\pm 0.065}$ | $0.519_{\pm 0.014}$ | 0.031 |

Table 5: Stress values for the `cities` task at the RC site. The highest-scoring manifold is always a spherical one.

| Model | Acc | Manifold $-\log S$ | | | |
|---|---|---|---|---|---|
| | | cylinder | flat | geodesic | sphere |
| Llama-3.2-3B-it | 0.549 | 2.071 | 1.931 | 2.118 | **2.285** |
| Qwen2.5-3B-it | 0.510 | 1.906 | 1.768 | 1.975 | **2.135** |
| gemma-2-2b-it | 0.493 | 2.070 | 1.947 | 2.073 | **2.248** |

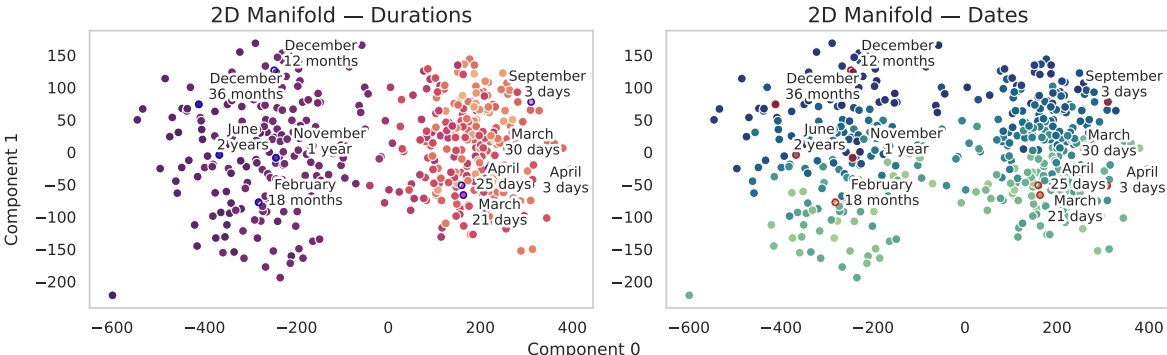

Figure 10: Multidimensional manifold constructed by Llama-3.2-3B-Instruct on the `duration` task. Component 0 is proportional to the duration, component 1 to the day of the year.

## 5.4 Generalizing SMDS Across Domains and Feature Types

Defining manifolds through distance functions enables extending SMDS beyond the mono-dimensional case. This sections provides two such examples.

**2D Manifold** We analyze the `duration` task in more detail as each sentence contains two temporal expressions. We use both the duration and starting day to define a 2D label, and run SMDS with the `linear` hypothesis as it is the only distance function supporting multidimensional labels. Doing so reveals a 2D manifold that displays properties from both features. Table 4 presents Wilcoxon $p$-values obtained by comparing the model to a control task, showing they are significant across models. We hypothesize the creation of such manifolds happens during the information flow discussed earlier: features are retrieved from multiple locations and combined. The recovered manifold is shown in Figure 10.

**Spatial Reasoning Domain** To demonstrate the versatility of SMDS beyond temporal reasoning, we apply it to a task grounded in geographic knowledge. We construct a dataset of prompts referencing various cities around the world and use their latitude and longitude to compute pairwise distances and reconstruct a manifold. While Gurnee & Tegmark (2023) demonstrates that geographic location is decodable from LMs' hidden states, their analysis is limited to a planar projection. We extend this by evaluating spherical, cylindrical, and geodesic-based geometries, and find that a spherical manifold best captures the structure of the representations. This again highlights how feature manifolds align with the true geometry of the underlying domain. Further details are provided in Appendix D.6.

## 6 Discussion & Conclusion

This paper introduces SMDS, a model-agnostic method for identifying and testing subspaces with a specified geometry via a linear projection. SMDS is not intended as a replacement for existing dimensionality reduction techniques; rather, it addresses a specific gap in their use: the ability to isolate subspaces in which representations conform to a hypothesized geometric structure. In SMDS, the target structure is specified through a user-defined distance function, in contrast to methods such as LDA or PCA where geometric patterns arise indirectly from the optimization objective. As a result, the method tests candidate geometric hypotheses rather than discovering entirely new ones, and its effectiveness depends on the quality and breadth of the hypotheses considered. By evaluating a sufficiently broad set of candidate structures, however, SMDS can help narrow down plausible manifold hypotheses. As a supervised approach, SMDS additionally requires labeled data and the ability to define a meaningful geometric structure over those labels. Finally, as with other supervised methods, care must be taken to mitigate overfitting, for example through appropriate regularization and a well-motivated selection of candidate geometries.

Our study establishes a connection between the geometry of representation manifolds and the causal language modeling process, demonstrating that a structured organization of knowledge is not only present but beneficial for model reasoning. By analyzing the persistence of these structures across tokens—particularly from the injection point to the answer—we provide compelling evidence that feature binding operates through continuous, task-relevant manifolds in the latent space. The persistence of manifolds across tokens suggests that language models transfer not just vectors, but structured representations, reinforcing the presence of a binding mechanism and extending prior evidence to more diverse tasks (Dai et al., 2024).

Although our experiments center on temporal reasoning, the proposed method extends to any task involving structured features on which a distance function can be defined, as we demonstrate in §D.6. Starting from hypothesis manifolds inspired by prior work, we obtain consistent, interpretable results, effectively reframing manifold analysis as a model selection problem. A compelling direction for future research is understanding how individual features combine into multidimensional manifolds. While we present initial evidence of composition, more expressive manifold hypotheses could offer deeper insights. SMDS lays the foundation for such investigations.

Our stress metric often yields tightly clustered scores. The approach followed in this study is to drastically increase experimental observations via repetition or bootstrapping to hone in on a single leading manifold for each task. Future studies could adopt different approaches: the development of more discriminative metrics, the use of larger and more varied datasets, and complementary intervention experiments such as the one in §5.3. There is a final option, however, which is to reassess the assumption that a single preferential manifold exists. An intriguing hypothesis is that models instead adopt multiple equally valid representational geometries and dynamically select them based on task context. Failure to isolate a single best manifold should signal that new hypotheses must be formulated, potentially accounting for the coexistence of multiple representational geometries.

In the scope of model reasoning, hypothesis-driven manifold analysis can serve as a basis for several lines of future work. For instance, combining SMDS with circuit discovery (Conmy et al., 2023) could help identify which operations LMs use to transform information throughout reasoning. Another promising direction is model steering (Park et al., 2024b), where knowledge of feature manifolds could inform methods that leverage these structures directly. Finally, systematically studying the role of noise in feature manifolds across layers, and whether mitigating it improves reasoning, offers another rich line of inquiry.

In sum, *shape happens*. Our work lays the foundational ground for interpreting and comparing representations in LMs through geometric structures. This invites further exploration into how manifold shapes are formed, combined, functionally employed in downstream reasoning, and how knowing about them could improve existing models.

## Limitations

Our use of language models trained on predominantly English corpora introduces an inherent bias toward the cultural norms of the Anglosphere. This is reflected in several design choices: the reliance on the Gregorian calendar for date expressions; the selection of names that are tokenized as single units, which tends to privilege Anglo-American names; and assumptions about seasonal properties (e.g., associating December with cold weather), which implicitly expects the location to be a country in the northern hemisphere, with a temperate or continental climate. The high accuracy and well-formed manifolds observed in these settings can therefore be seen as indicators of such biases. SMDS could find use as a diagnostic tool, uncovering how underlying representations reflect these biases.

In our work, we omit fuzzy expressions for which it is not possible to define precise temporal pointers (e.g., "in the morning," "later," and "next week") and therefore an exact location on a feature manifold. As Kenneweg et al. (2025) show, fuzziness in a temporal expression is a key factor in performance degradation. Future works could better characterize the interplay between fuzziness in temporal expressions and the quality of feature manifolds.

## Acknowledgments

Federico Tiblias is supported by the Konrad Zuse School of Excellence in Learning and Intelligent Systems (ELIZA) through the DAAD programme Konrad Zuse Schools of Excellence in Artificial Intelligence, sponsored by the Federal Ministry of Education and Research.

This work has been funded by the LOEWE Distinguished Chair "Ubiquitous Knowledge Processing," LOEWE initiative, Hesse, Germany (Grant Number: LOEWE/4a//519/05/00.002(0002)/81), by the German Federal Ministry of Education and Research, and by the Hessian Ministry of Higher Education, Research, Science and the Arts within their joint support of the National Research Center for Applied Cybersecurity ATHENE.

We thank Alireza Bayat Makou, Andreas Waldis and Hovhannes Tamoyan (UKP Lab, TU Darmstadt) for their feedback on an early draft of this work. We also thank Marco Nurisso (Politecnico di Torino) and Anmol Goel (UKP Lab, TU Darmstadt) for the many insightful discussions that helped shape this paper.

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

## Contents

## A   Extended Literature Review

**Linear Representations & Feature Manifolds**   The linear representation hypothesis proposes that language models encode interpretable features as directions in their latent space, with concepts expressed as sparse linear combinations of these directions (Park et al., 2024b; Modell et al., 2025). Recent work extends this view, revealing that related features tend to organize into structured manifolds. For example, ring-like structures (Engels et al., 2025), logarithmic progressions (AlquBoj et al., 2025), U-shaped curves (Engels et al., 2025; Modell et al., 2025), clusters organized around the vertices of geometric polytopes (Park et al., 2024a), and higher-dimensional surfaces (Gurnee & Tegmark, 2023). Other studies on multilingual LMs have also investigated structures in the latent space and consistently found shared representations across languages (Peng & Søgaard, 2024; Artetxe et al., 2020; Chang et al., 2022; Conneau et al., 2020, *inter alia*). Beyond static feature geometry, recent work has shown that such linear and manifold structures also underpin model behaviors, including truthfulness (Marks & Tegmark, 2024) and the interaction between reasoning and memorization (Hong et al., 2025).

**Existing Dimension Reduction Methods**   Several linear dimensionality reduction techniques have been applied to recover structure from language model representations: Principal Component Analysis (PCA) identifies directions of maximal variance in the embedding space (Gurnee & Tegmark, 2023; Modell et al., 2025); Linear Discriminant Analysis (LDA) finds directions that best separate labeled categories (Park et al., 2024a); Partial Least Squares Regression (PLS) identifies components that most strongly covary with target labels (Wold et al., 2001; El-Shangiti et al., 2025; Heinzerling & Inui, 2024); and Multi-Dimensional Scaling (MDS) seeks low-dimensional embeddings that preserve pairwise distances from the original space (Marjieh et al., 2025). In addition to these linear methods, some non-linear techniques such as t-SNE and UMAP have also been applied (van der Maaten & Hinton, 2008; Healy & McInnes, 2024; Subhash et al., 2023). Other dimensionality reduction techniques may also yield promising insights, but to our knowledge have not yet been systematically applied to probing LM representations (Trofimov et al., 2022; Tulchinskii et al., 2025).

Probes have become a widely used tool for analyzing the internal representations of language models. Typically, a probe is a simple classifier trained to predict a specific linguistic or conceptual property from a model's hidden states. They have been employed to study a range of linguistic features, including morphology and syntax (Belinkov et al., 2017; Hewitt & Manning, 2019), as well as broader aspects of neural network behavior (Alain & Bengio, 2017; Jin et al., 2025). Other works has used probes to examine the sparsity of feature representations (Gurnee et al., 2023), detect model truthfulness (Li et al., 2023; Marks & Tegmark, 2024), and uncover the representations of concepts such as world locations, temporal quantities and numbers (Gurnee & Tegmark, 2023; Engels et al., 2025; Levy & Geva, 2025). However, probe-based interpretations remain debated as probe performance does not necessarily imply mechanistic use (Jawahar et al., 2019; Tenney et al., 2019; Niu et al., 2022), a limitation less applicable to representation-geometry approaches that incorporate causal interventions (Heinzerling & Inui, 2024).

Another prominent technique used in interpretability works is the Sparse Auto Encoder (SAE), a neural network building a mapping from the dense activation space of a LM to a high-dimensional, sparse, latent space such that single neurons of a SAE represent atomic concepts (Bricken et al., 2023; Huben et al., 2023). SAEs have been successful at recovering vast collections of monosemantic, interpretable features at scale (Templeton et al., 2024), but have also found usage in unlearning (Farrell et al., 2024), detecting internal causal graphs (Marks et al., 2024) and identifying circuits (Minegishi et al., 2024). Recent work has also explored post-hoc interpretation of SAE features themselves, for example through agentic explainer frameworks such as SAGE (Han et al., 2025). While SAEs have shown promise for LLM interpretability, they face substantial critiques and limitations that challenge their effectiveness and reliability. Representations identified by SAEs may fall victim of "feature absorption," complicating the disentanglement of atomic features (Chanin et al., 2025). In model steering, simple baselines have been observed outperforming SAEs (Wu et al., 2025; Kantamneni et al., 2025). Lastly, SAEs are expensive to construct as they require extremely large dimension of their latent space and necessitate in some cases billions of tokens for training. Their construction also makes them model-specific, preventing transferability (Sharkey et al., 2025).

In this work, we primarily compare SMDS with other linear dimensionality reduction techniques. While non-linear methods and SAEs may offer valuable insights, we do not focus on them here due to their high computational demands. Our goal is to enable a scalable and systematic exploration of feature manifolds across models and tasks. This requires lightweight methods that can be efficiently applied in closed form, making linear approaches better suited to the scope of our investigation.

**Temporal Reasoning**   Temporal reasoning refers to the ability to interpret and manipulate expressions that describe temporal information[3] in order to determine when events occur or how they relate temporally (Jia et al., 2018a). Such reasoning tasks often require composing multiple temporal expressions to answer nuanced, time-sensitive questions.

Several datasets exist that seek to benchmark LMs across different facets of temporal reasoning. Some evaluate factual recall over time (Jia et al., 2018b; Chen et al., 2021; Jia et al., 2021), others focus on temporal understanding of real-world scenarios (Zhou et al., 2019; Fatemi et al., 2025), and yet others probe the temporal arithmetic capabilities of LMs (Tan et al., 2023). Lastly, some works have aggregated existing benchmarks in order to evaluate broader capabilities such as symbolic, commonsense, and event reasoning (Wang & Zhao, 2024; Chu et al., 2024).

Existing benchmarks primarily assess overall performance on complex tasks involving multiple temporal expressions and reasoning types. Simpler tasks focusing on specific types of temporal expressions, despite being foundational to temporal understanding, remain underexplored. To enable a mechanistic investigation of how language models process temporal information, homogeneous datasets that isolate specific facets of temporal expressions are required.

---

[3]For example, dates (e.g., "May 1, 2010"), times ("9 pm"), or temporal relations ("before," "in the morning").

Table 6: Possible temporal expressions for each task.

| Dataset | Temporal expression set |
|---------|------------------------|
| `cities` | Uniformly sampled based on location from the World Cities Database, considering only prominent cities or cities with $> 10.000$ inhabitants for US and Canada. |
| `date`, `date_season`, `date_temperature` | Uniformly sampled from all 365 days of a non-leap year. |
| `duration` | Dates sampled in the same way as `date`, durations uniformly sampled from fixed set: 1 day, 2 days, 3 days, 4 days, 5 days, 6 days, 7 days, 8 days, 9 days, 10 days, 1 week, 2 weeks, 3 weeks, 4 weeks, 7 days, 10 days, 14 days, 21 days, 25 days, 30 days, 1 month, 2 months, 3 months, 4 months, 6 months, 8 months, 4 weeks, 6 weeks, 8 weeks, 10 weeks, 1 year, 2 years, 3 years, 4 years, 12 months, 18 months, 24 months, 36 months. |
| `notable` | Uniformly sampled from a fixed set, extracted from Wikipedia. Omitted from brevity, full dataset available in the code repository. |
| `time_of_day`, `time_of_day_phase` | Action time uniformly sampled from all hours at :00, :15, :30, :45. Reference time sampled uniformly from all times of the day. |

## B  Temporal Taxonomy & Datasets

This section describes the synthetic datasets we have generated to probe atomic aspects of temporal understanding.

**Taxonomy**  Various annotation schemes have been developed to characterize temporal expressions such as TIMEX3 (Pustejovsky et al., 2010), TIMEX2 (Ferro et al., 2003), TIMEX (Setzer, 2001) and TimeML (Saurí et al., 2006), as well as several variants. We take inspiration from TIMEX1-3 to construct several synthetic datasets. Each one covers a specific family of temporal expressions (Table 2):

- `date`: Refers to a specific calendar date. To explore periodic reasoning, we omit the year;

- `time_of_day`: Specifies a precise moment in the day;

- `duration`: Defines a duration and its starting point;

- `periodic`: Refers to events that recur with a given frequency;

- `notable`: Contains an indirect but precise reference to an event taking place in a given moment in time.

The taxonomy has been defined in such a way that temporal expressions have a unique, precisely-defined associated numerical quantity. We have chosen to omit fuzzy expressions for which it is not possible to define precise temporal pointers (e.g. "in the morning", "later", "next week") and therefore an exact position in a feature manifold.

**Dataset Creation**  We build each sentence in the dataset by combining three contextual sentences and a termination that elicits reasoning. Each sentence contains a name, action and temporal expressions which are all uniformly sampled from a given set. Names and actions have been generated via ChatGPT and checked manually to be consistently formatted and the resulting sentences grammatically correct. Names have been chosen so that they are not broken up into separate tokens. Temporal expressions of the `notable` task have been obtained from Wikipedia[4] and have been rewritten via ChatGPT and checked manually to ensure consistence. For all datasets, each sentence contains exactly one temporal expression. We chose not to

---

[4]https://en.wikipedia.org/wiki/Timeline_of_the_20th_century

Table 7: Additional examples for each task.

| Dataset | # Samples | Examples |
|---|---|---|
| cities | 2000 | Luke lives in Boston. William lives in Toronto. Michael lives in Cancún. The person who lives closest to Luke is |
| | | Mark lives in Leuven. Jack lives in Heidelberg. Dallas lives in Messina. The person who lives closest to Mark is |
| date | 1992 | Brandon donated clothes on the 29th of September. Bob donated clothes on the 31st of August. Jerry donated clothes on the 27th of September. The first person that donated clothes was |
| | | Matt visited a new city on the 22nd of February. Josh visited a new city on the 14th of February. Frank visited a new city on the 1st of March. The first person that visited a new city was |
| date_season | 2000 | Emily mowed the lawn on the 8th of December. Blake mowed the lawn on the 30th of April. Walker mowed the lawn on the 27th of June. The only person that mowed the lawn in fall is |
| | | Rose painted a mural on the 16th of June. Robert painted a mural on the 13th of July. Martin painted a mural on the 27th of July. The only person that painted a mural in spring is |
| date_temperature | 2000 | Richard left for vacation on the 25th of June. Neil left for vacation on the 22nd of December. April left for vacation on the 22nd of August. The only person that left for vacation in a cold month is |
| | | Jason returned from vacation on the 12th of February. Connor returned from vacation on the 21st of March. Rachel returned from vacation on the 19th of October. The only person that returned from vacation in a warm month is |
| duration | 3000 | Maria is starting their internship on the 15th of December and is set to run for 25 days. George is starting their internship on the 13th of December and is set to run for 14 days. Laura is starting their internship on the 3rd of December and is set to run for 1 week. The person whose internship ends first is |
| | | Hunter runs a festival booth on the 27th of December staying open for 10 days. George runs a festival booth on the 12th of November staying open for 9 days. Connor runs a festival booth on the 20th of December staying open for 9 days. The person whose festival booth ends first is |
| notable | 2000 | Robert was born on the day the MV Doña Paz sank. Maria was born on the day the independent State of Palestine was proclaimed. Andrew was born on the day the Dayton Accords were signed. The oldest is |
| | | Neil was born on the day Herbert Hoover was inaugurated as President. Leon was born on the day James Joyce published Ulysses. Alice was born on the day Mandatory Palestine was established. The oldest is |
| time_of_day | 3000 | Steve watches a movie at 23:15. April watches a movie at 11:45. Charlie watches a movie at 7:15. It is now 2:58. The last person who watched a movie is |
| | | Charlie watches TV at 4:45. Richard watches TV at 12:15. Steve watches TV at 4:30. It is now 14:42. The last person who watched TV is |
| time_of_day_phase | 2000 | Leon goes for a walk at 6:15. Brandon goes for a walk at 18:30. Matt goes for a walk at 5:00. The only person that goes for a walk in the morning is |
| | | John writes in a journal at 4:45. Matt writes in a journal at 5:00. Luke writes in a journal at 20:45. The only person that writes in a journal in the evening is |

include more, using composite expressions, so as to obtain cleaner feature manifolds. The only exceptions are the `duration` and `time_of_day` datasets that contains two. This was necessary in order to formulate non-trivial questions that require reasoning across time spans. The `notable` task not only requires comparing different expressions but also involves factual recall of events from parametric memory. Variants `date_season`, `date_temperature`, and `time_of_day_phase` contain the same contextual sentences as the original tasks but a different termination that elicits a classification-based form of reasoning. Finally, we note that the number of examples per dataset varies. This is necessary to ensure that, after filtering for correctly answered instances, a sufficient number of activations remain for SMDS training. For consistency, we require a minimum of 500 correctly classified examples per model-task pair and cap the number of activations used in manifold search at this threshold. See Table 6 and Table 7 for a more extensive collection of templates and examples.

Table 8: BERTScore-based variability statistics for our dataset compared to prior datasets commonly used in studies of representational geometry and the linear representation hypothesis. Our dataset exhibits substantially higher variability across similarity pairs.

| Dataset | Mean Similarity | Std |
|---|---|---|
| *Our datasets* | | |
| cities_3way | 0.8170 | 0.0217 |
| date_3way_season | 0.8306 | 0.0244 |
| date_3way_temperature | 0.8480 | 0.0240 |
| date_3way | 0.8218 | 0.0250 |
| duration_3way | 0.7975 | 0.0311 |
| notable_3way | 0.8099 | 0.0281 |
| periodic_3way | 0.8023 | 0.0302 |
| time_of_day_3way_phase | 0.8169 | 0.0265 |
| time_of_day_3way | 0.8174 | 0.0257 |
| *Heinzerling & Inui (2024)* and El-Shangiti et al. (2025) | | |
| P569 birthyear | 0.8582 | 0.0444 |
| P570 death year | 0.8485 | 0.0517 |
| P625.lat latitude | 0.8289 | 0.0489 |
| P625.long longitude | 0.8598 | 0.0409 |
| P1082 population | 0.8544 | 0.0430 |
| P2044 elevation | 0.8410 | 0.0445 |
| *Engels et al. (2025)* | | |
| days of week | 0.9627 | 0.0171 |
| month of year | 0.9617 | 0.0144 |

## B.1 Data Variability

Overall, as shown in Table 8, the dataset we created demonstrates higher variability compared to datasets curated by prior work in related areas. In particular, we compute several BERTScore-based (Zhang et al., 2019) variability metrics across all splits of our dataset and compare them against the dataset used in Heinzerling & Inui (2024) and El-Shangiti et al. (2025), as well as the dataset in Engels et al. (2025), works that established the study of representational geometry and the linear representation hypothesis. We measure variability by computing BERTScore F1 for every possible pair of texts in a dataset split, using the model's predicted-reference pairs formed via 5000 randomly sampled combinations. The mean similarity is the average of these pairwise F1 scores, while the std is the sample standard deviation of the same set of scores.

## C  Supervised Multi-Dimensional Scaling

In this section we provide further details on the dimensionality reduction method we use throughout the paper, as well as highlight its differences and similarities with other techniques which served as inspiration.

**Description**  SMDS is based on the assumption that points $X \in \mathbb{R}^{n \times d}$ in the residual stream roughly lie on a feature manifold that can be parametrized with labels $y \in Y$ with $y_i \in [0, 1]$. Distances on this ideal feature manifold are assumed similar to Euclidean distances in the residual stream. Formally, given activations $X \in \mathbb{R}^{n \times d}$, two samples $x_i, x_j \in X$ and a linear projection $W \in \mathbb{R}^{m \times d}$ from the full space to the manifold subspace, we assume that $d(y_i, y_j) \approx \|W(x_i - x_j)\|$. To find $W$, we can minimize Eq. 1, reported here for readability:

$$\mathcal{L} = \sum_{i<j} \left( \|W(x_i - x_j)\|^2 - d(y_i, y_j)^2 \right)^2.$$

The problem is solved as follows. First, ideal distances $d(y_i, y_j)$ between labels are computed and the squared distance matrix is defined as:

$$D_{ij} \coloneqq d(y_i, y_j)^2. \tag{4}$$

Then, classical MDS is performed. Double centering is applied:

$$H \coloneqq I - \frac{1}{n} \mathbf{1} \mathbf{1}^\top, \quad B \coloneqq -\frac{1}{2} HDH. \tag{5}$$

$B$ is eigen-decomposed and a low-dimensional embedding $Y$ is obtained:

$$B = V \Lambda V^\top, \tag{6}$$

$$Y \coloneqq V_m \Lambda_m^{1/2} \in \mathbb{R}^{n \times m}, \tag{7}$$

with $V_m$ the top $m$ eigenvectors and $\Lambda_m$ the corresponding eigenvalues. The embeddings $Y$ represent the locations of data points in the parametrized approximation of the manifold such that $\|Y_i - Y_j\| \approx d(y_i, y_j)$. The following steps perform regression to find a mapping from datapoints $X$ to this subspace. We center $X$ and $Y$ by subtracting their mean:

$$X_c = X - \overline{X}, \quad Y_c = Y - \overline{Y}. \tag{8}$$

Substituting in Eq. 1:

$$\|X_c W^\top W X_c^\top - Y_c Y_c^\top\|^2. \tag{9}$$

At the optimum $W$ we get that $X_c W^\top \approx Y_c$. Solving Eq. 1 is expensive, however we can approximate $W$ by solving a proxy problem:

$$W = \arg\min_{\hat{W}} \|X_c \hat{W}^\top - Y_c\|. \tag{10}$$

this is the same formulation as a linear probe, but with $Y$ being computed from the label using MDS. A solution is easily found:

$$W = Y_c^\top X_c \left( X_c^\top X_c \right)^{-1}. \tag{11}$$

A regularization term can be added to Eq.11 to make the resulting projection more robust:

$$W = Y^\top X_c \left( X_c^\top X_c + \alpha I \right)^{-1}. \tag{12}$$

In all our experiments, we set $\alpha = 0.1$.

The embeddings $Y$ represent the locations of data points in the parametrized approximation of the manifold (Figure 2). In principle, if such embeddings are already known, the preceding steps can be skipped entirely. Computing arbitrary $d(y_i, y_j)$ just gives more flexibility. By using SMDS to perform a search across several candidate hypotheses, the problem of identifying a manifold can be reduced to one of model selection: one only needs to perform SMDS on several metrics $d(y_i, y_j)$ or parametrized manifolds $Y$, and compare them using a quality metric like stress.

**Comparison to other methods** SMDS is an extension of MDS and uses it as part of the procedure. There is, however, a key difference in its use case: while classical MDS is unsupervised and only learns a lower-dimensional mapping that preserves Euclidean distances, SMDS first builds a distance matrix from labels and then uses it to learn the actual projection via regression. There are also differences in the stress metric we use (Eq. 3): classical normalized stress (Amorim et al., 2014) evaluates the error between distances in the original and lower-dimensional space; our formulation effectively does the same, but between the projected and the ideal subspace $Y$.

The first term in Eq. 1 can be reformulated:

$$\|W(x_i - x_j)\|^2 = (W(x_i - x_j))^\top (W(x_i - x_j)),$$
$$= (x_i - x_j)^\top W^\top W(x_i - x_j),$$
$$= (x_i - x_j)^\top M(x_i - x_j),$$

with $M \in \mathbb{R}^{n \times n}$ being a positive semi-definite matrix. This is the squared Mahalanobis distance, widely used in Distance Metric Learning. In fact, many other dimensionality reduction techniques can be described as Distance Metric Learning algorithms (Suárez-Díaz et al., 2020).

SMDS is closely related to probes, which have been extensively used in prior works (Belinkov, 2022; Li et al., 2022; Gurnee & Tegmark, 2023, *inter alia*). Some have successfully employed circular probes to recover feature manifolds (Engels et al., 2025) and study other cyclical patterns such as number encodings (Levy & Geva, 2025), but to the best of our knowledge no prior works have used MDS to build probes of arbitrary shape.

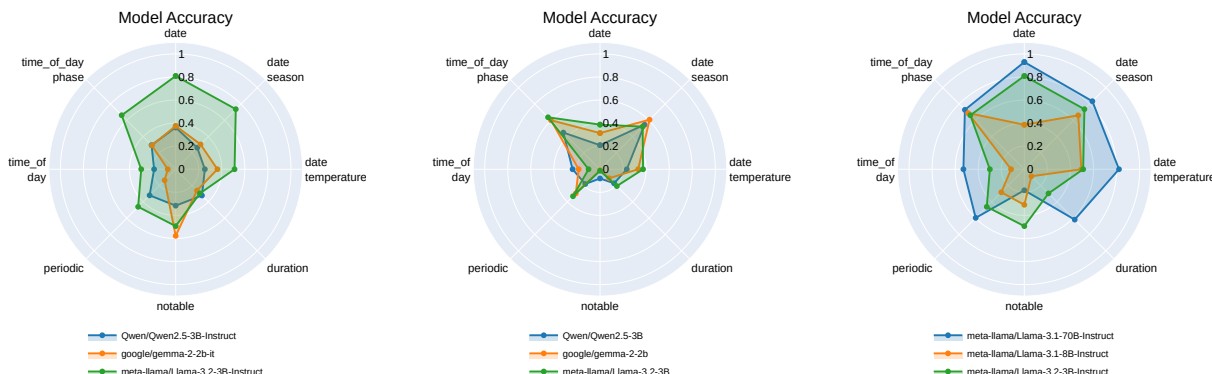

Figure 11: Accuracy on temporal tasks. Accuracy is low across the board, with only Llama models achieving above-chance accuracy. The 3B Llama variant is also shown outperforming the 8B one.

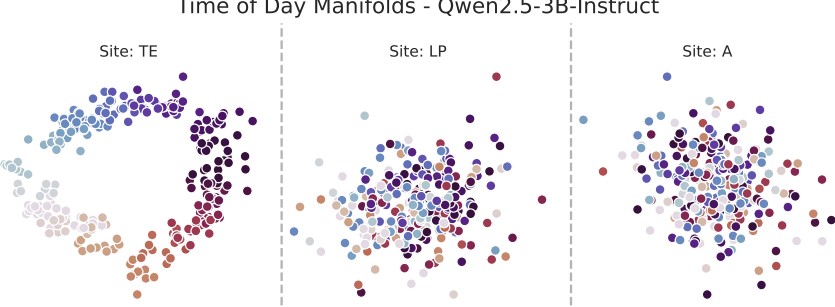

Figure 12: Circular manifolds on the `time_of_day` task. SMDS cannot find any structure on LP and A despite one being present at the TE site.

# D    Supplementary Experiments

## D.1    Model Performance

We evaluate exact match accuracy across all tasks and models, finding that performance is generally very low, with only instruction-tuned models from the Llama family outperforming random chance. This is notable because, despite poor task performance, the models still produce well-defined feature manifolds. Among Llama models, we find the more recent Llama-3.2-3B-Instruct outperforms its 8B counterpart, while the 70B version displays stronger performance in almost all tasks (Figure 11).

While stress is a good indicator of performance, as discussed in Appendix 5.3, we observe no significant correlation for the three models of the main analysis (Spearman's $\rho = -0.034$). We hypothesize that while most LMs effectively structure knowledge internally, some struggle to leverage it during generation. This might explain their lower performance. Another possibility is that the specific wording of the prompt does not allow LMs to effectively recover information from context. In that case, chain-of-thought prompting (Wei et al., 2022) may improve performance.

## D.2    Additional Observations on Manifold Analysis

Manifold analysis reveals consistent patterns across model scales. As Figure 14 shows, larger models from the same family tend to converge to the same internal representations as their smaller counterparts. This suggests architecture and pre-training data play a pivotal role in determining LMs representations.

We observe two instances where manifold analysis exhibits unexpected behaviors. On the `time_of_day` task, SMDS is unable to recover well-organized manifolds at the LP site despite a clear, preferential circular manifold

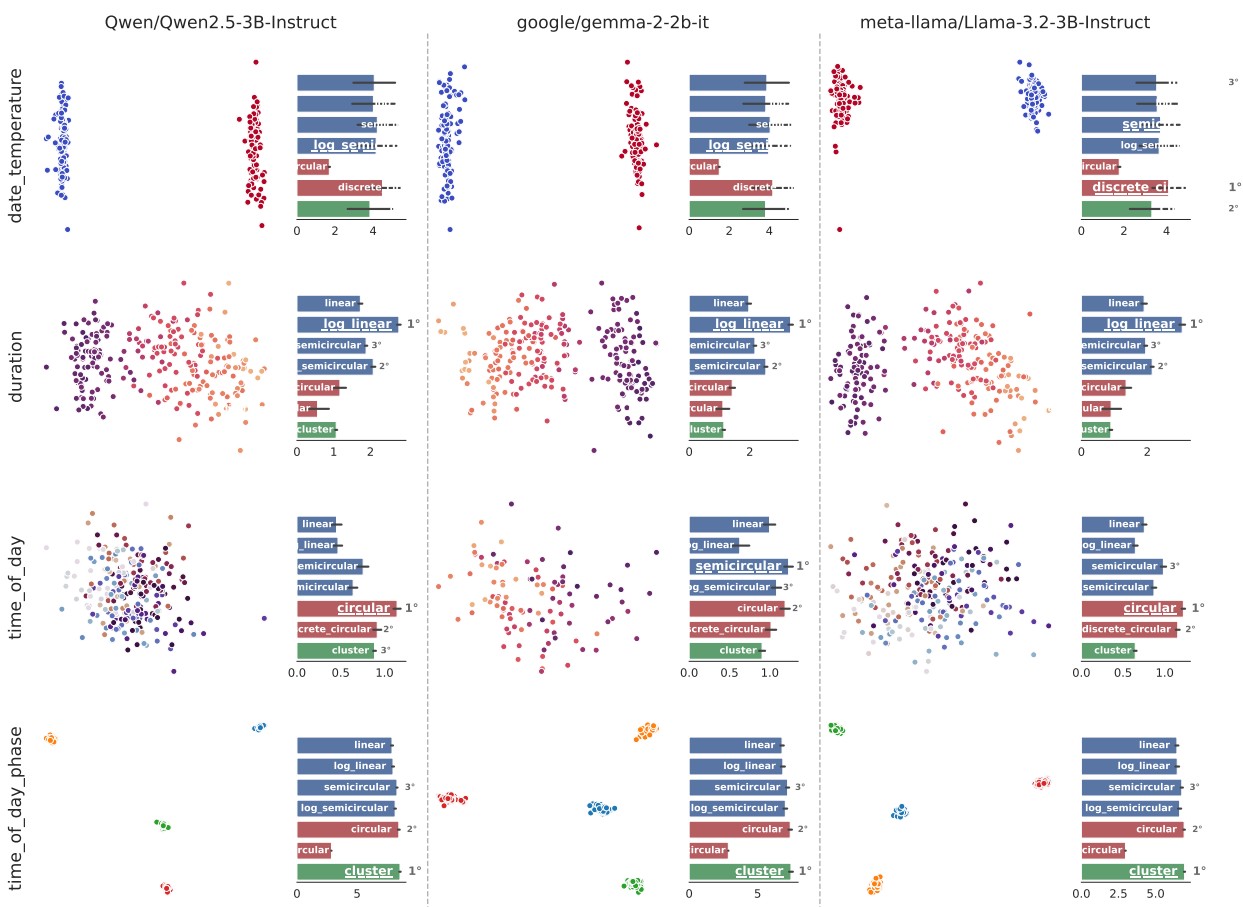

Figure 13: Additional manifolds for different tasks and models. Continued from Figure 3; error bars are shown in black. Manifold topology: ■ linear; ■ cyclical; ■ categorical;

being present at the TE site (Figure 12). The lack of transferability between the two sites can be explained by noting that `time_of_day` sentences contain two temporal expressions in the same format instead of one. The two representations may interfere destructively, preventing their recovery. When also considering findings from §D.5, it is also possible that, for this specific prompt, the LP site is not storing any semantically relevant information. Future works could start from tasks such as this to characterize how multiple feature manifolds combine, and in which token is this information encoded.

On the `date_temperature` task (Figure 13), the clusters are correctly identified but the scoring yields unreliable values. This is expected when considering how distances are computed in the binary cluster scenario: two clusters can be modelled correctly by any hypothesis manifold, as there is no order that can be enforced. This signals caution and suggests reverting to simpler probes to evaluate binary features.

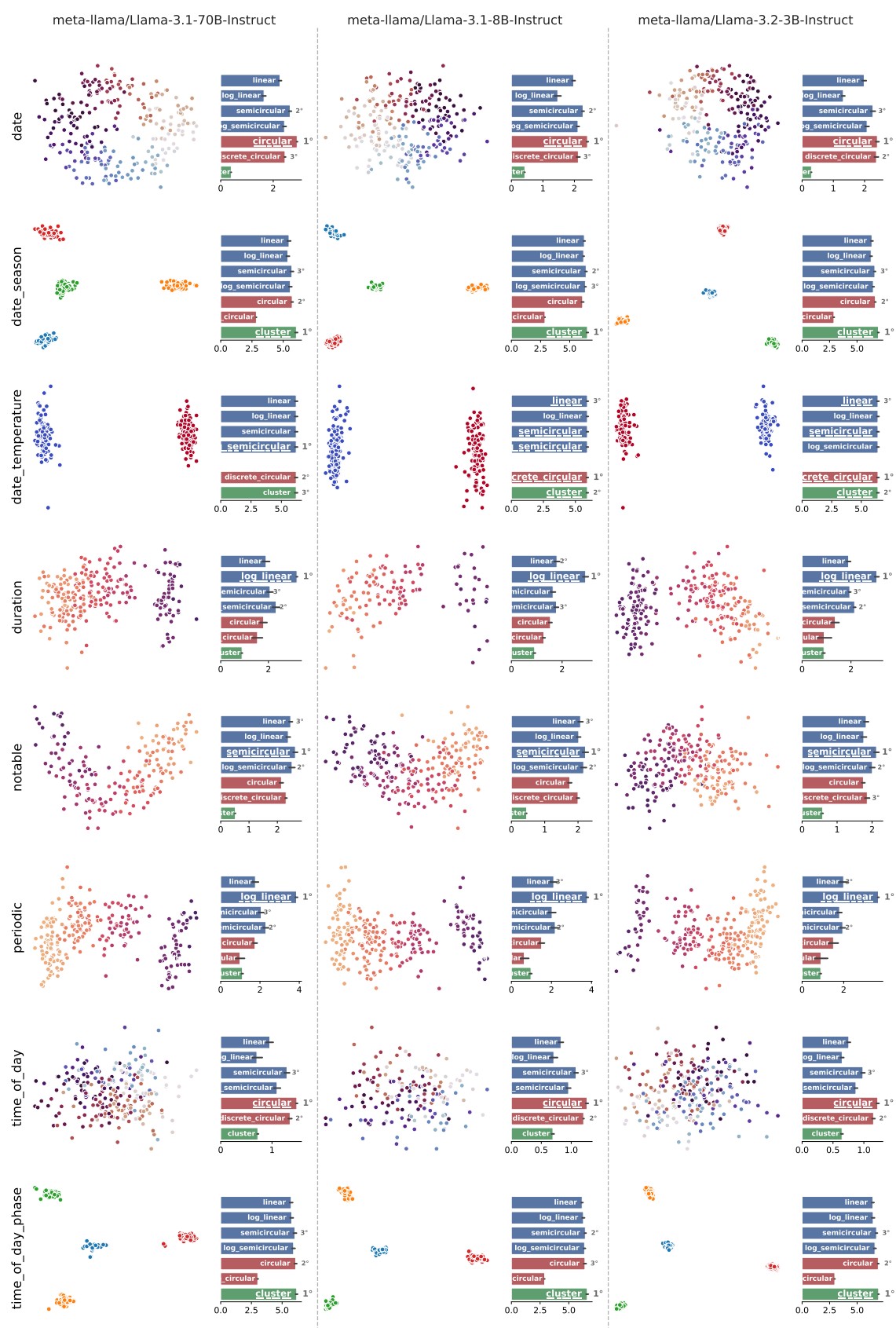

Figure 14: Feature manifolds for models at different sizes. The preferential manifold is consistent across scales. Error bars are shown in black. Manifold topology: ■ linear; ■ cyclical; ■ categorical;

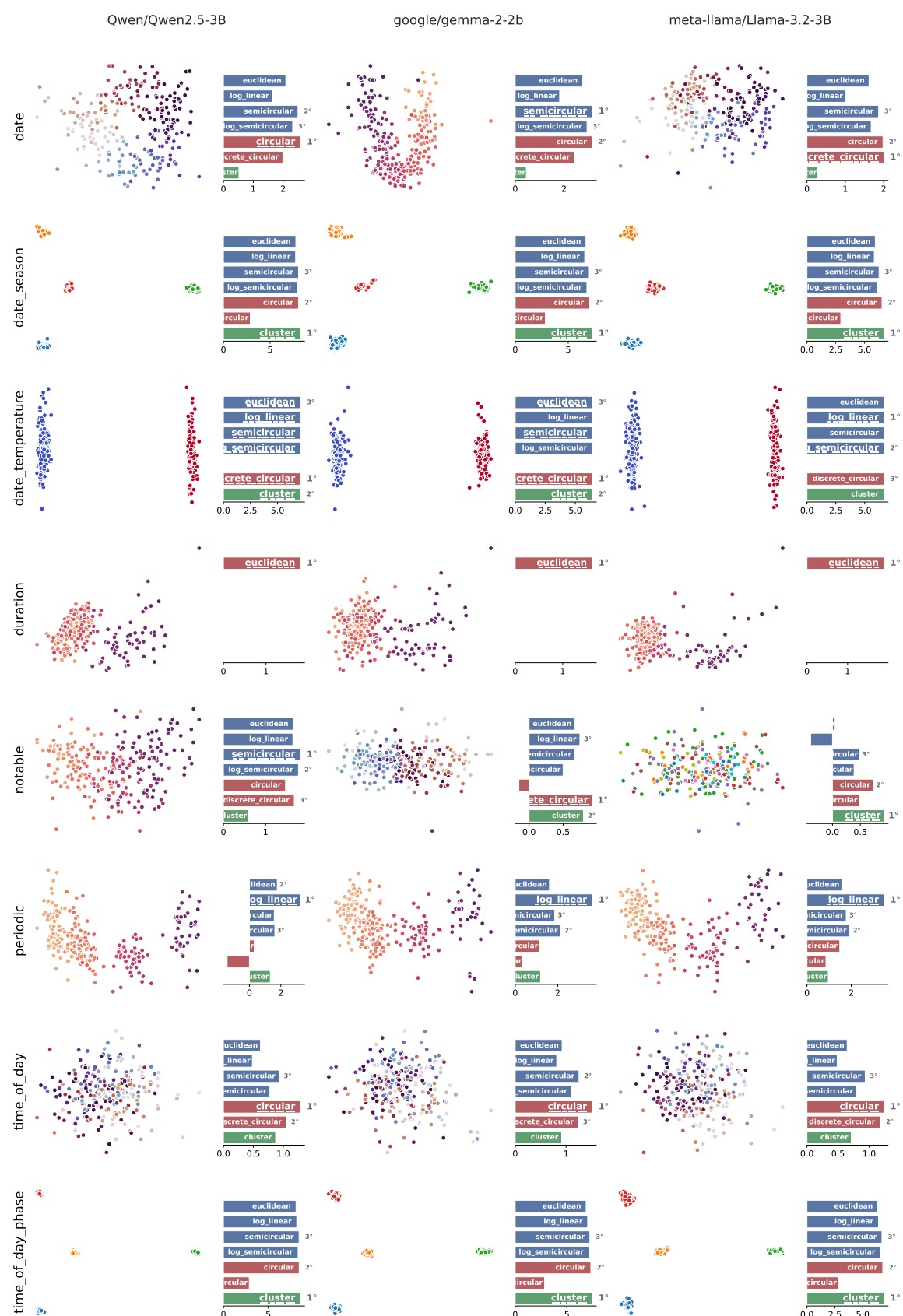

Figure 15: Feature manifolds for base models. Geometries are consistent with the instruction-tuned counterparts in most cases. Manifold topology: ■ linear; ■ cyclical; ■ categorical;

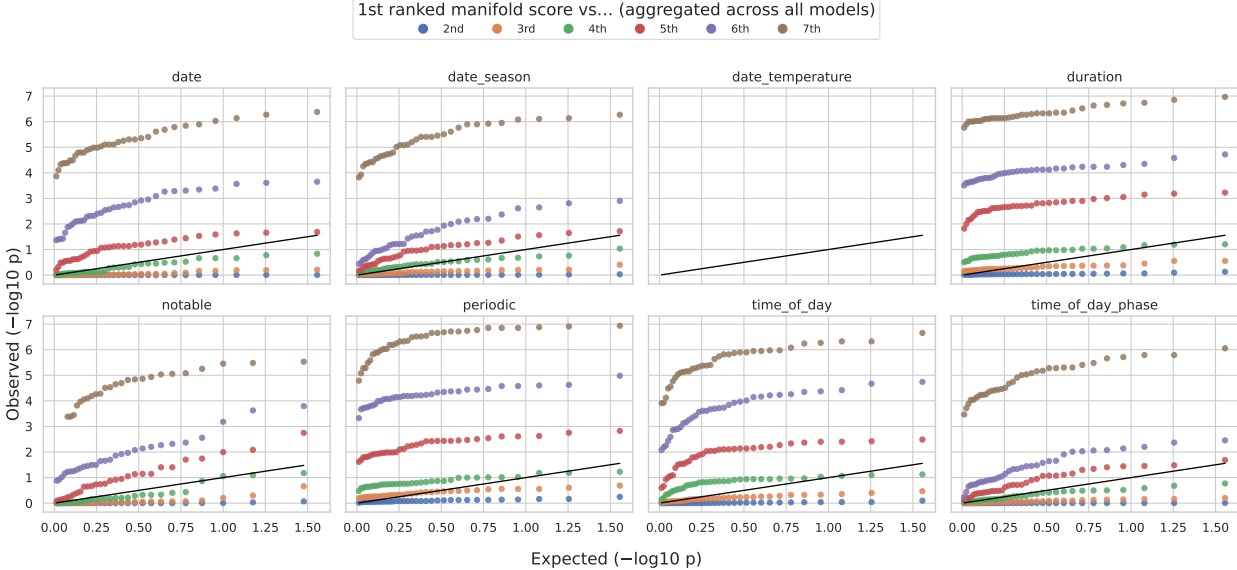

Figure 16: Q-Q plot of $p$-values obtained from a Nemenyi test comparing the stress scores of the first ranked manifold with all others for a given dataset. For all datasets except for `date_temperature`, the best identified manifold scores significantly higher than the alternative starting, on average, from the 3th-4th one.

## D.3 Establishing Statistical Significance

SMDS tends to return tightly clustered stress scores across different hypotheses, making it hard to identify a single best manifold. To overcome this issue, we resort to statistical testing to isolate a manifold that is statistically better than the alternatives.

First, we turn to a 10-fold cross-validation setup with 5 repetitions in which SMDS is trained on 9 folds and evaluated on the 10th. We group observation across dataset, model, and rank of the manifold, and perform a Friedman test. For all groups that achieve statistical significance ($p < 0.05$) we perform a post-hoc Nemenyi test to evaluate the significance of manifolds ranks on a given dataset. The resulting Q-Q plot in Figure 16 shows that for most tasks the manifolds ranked 1st performs comparably to manifolds ranked 2nd to 4th, and significantly better than the rest. For `duration`, `periodic`, and `time_of_day` this further reduces to manifolds 2nd to 3rd. The binary nature of `date_temperature` produces very homogeneous scores across all datasets and therefore we are not able to verify any significance. Results are summarized in Table 3, showing the best-scoring manifold for each task and model.

To further separate results, we employ bootstrapping. We run 500 independent bootstrap iterations, each time training on a dataset obtained by sampling with replacement from the original data and evaluating performance on the corresponding out-of-bag samples. We perform Friedman and Nemenyi tests as before and finally draw critical difference diagrams across all datasets. We perform this analysis over all models save for Llama-3.1-70B-IT which we exclude due to compute limitations. As Table 9 shows, confidence intervals become very narrow and thus provide statistically significant results for all models and tasks. Figures 5, 17 show the overall ranking of manifolds across tasks confirming the results obtained in the previous analysis and establishing a clear best-ranking hypothesis for each task, the only exception once again being the `date_temperature` dataset.

Taken together, the repeated cross-validation and bootstrap analyses provide consistent evidence that our conclusions are stable across data splits. Across both settings, statistical tests yield convergent significance patterns and identify the same top hypotheses for each task (with the expected exception of `date_temperature`, where scores are inherently homogeneous). This agreement across two complementary resampling protocols supports the claim that SMDS manifold selection is robust to variation in the underlying data split.

Table 9: Manifold scores for each model and dataset in a bootstrap setting with 500 iterations. Standard error shown in grey, best manifold bolded.

| Dataset / Manifold | date $-\log S$ | date_season $-\log S$ | date_temperature $-\log S$ | duration $-\log S$ | notable $-\log S$ | periodic $-\log S$ | time_of_day $-\log S$ | time_of_day_phase $-\log S$ |
|---|---|---|---|---|---|---|---|---|
| *Llama-3.1-8B-IT* | | | | | | | | |
| lin | $1.733_{\pm 0.005}$ | $6.128_{\pm 0.007}$ | $\mathbf{5.703}_{\pm 0.007}$ | $1.626_{\pm 0.008}$ | $1.912_{\pm 0.004}$ | $2.137_{\pm 0.009}$ | $0.837_{\pm 0.003}$ | $6.051_{\pm 0.007}$ |
| log_lin | $1.310_{\pm 0.006}$ | $6.060_{\pm 0.007}$ | $5.703_{\pm 0.007}$ | $\mathbf{2.708}_{\pm 0.007}$ | $1.849_{\pm 0.005}$ | $\mathbf{3.696}_{\pm 0.005}$ | $0.679_{\pm 0.004}$ | $6.132_{\pm 0.007}$ |
| semic | $2.035_{\pm 0.005}$ | $6.288_{\pm 0.006}$ | $\mathbf{5.703}_{\pm 0.007}$ | $1.647_{\pm 0.007}$ | $\mathbf{2.071}_{\pm 0.004}$ | $2.250_{\pm 0.009}$ | $1.075_{\pm 0.003}$ | $6.353_{\pm 0.006}$ |
| log_semic | $1.879_{\pm 0.005}$ | $6.217_{\pm 0.007}$ | $5.703_{\pm 0.007}$ | $1.796_{\pm 0.008}$ | $2.013_{\pm 0.004}$ | $2.487_{\pm 0.010}$ | $0.959_{\pm 0.003}$ | $6.260_{\pm 0.007}$ |
| circ | $\mathbf{2.210}_{\pm 0.004}$ | $5.981_{\pm 0.005}$ | $0.000_{\pm 0.000}$ | $1.575_{\pm 0.008}$ | $1.669_{\pm 0.003}$ | $1.893_{\pm 0.009}$ | $\mathbf{1.248}_{\pm 0.002}$ | $6.365_{\pm 0.006}$ |
| disc_circ | $2.032_{\pm 0.003}$ | $3.048_{\pm 0.002}$ | $\mathbf{5.703}_{\pm 0.007}$ | $1.349_{\pm 0.014}$ | $1.876_{\pm 0.003}$ | $1.377_{\pm 0.013}$ | $1.178_{\pm 0.002}$ | $3.032_{\pm 0.002}$ |
| clust | $0.459_{\pm 0.002}$ | $\mathbf{6.421}_{\pm 0.005}$ | $\mathbf{5.703}_{\pm 0.007}$ | $1.105_{\pm 0.004}$ | $0.619_{\pm 0.002}$ | $1.127_{\pm 0.002}$ | $0.886_{\pm 0.003}$ | $\mathbf{6.539}_{\pm 0.006}$ |
| *Llama-3.2-3B-IT* | | | | | | | | |
| lin | $1.854_{\pm 0.006}$ | $6.310_{\pm 0.006}$ | $6.306_{\pm 0.007}$ | $1.866_{\pm 0.005}$ | $1.808_{\pm 0.004}$ | $1.805_{\pm 0.009}$ | $0.720_{\pm 0.002}$ | $6.450_{\pm 0.007}$ |
| log_lin | $1.225_{\pm 0.006}$ | $6.226_{\pm 0.007}$ | $6.306_{\pm 0.007}$ | $\mathbf{2.943}_{\pm 0.004}$ | $1.743_{\pm 0.004}$ | $\mathbf{3.573}_{\pm 0.005}$ | $0.599_{\pm 0.003}$ | $6.463_{\pm 0.007}$ |
| semic | $2.146_{\pm 0.006}$ | $6.550_{\pm 0.006}$ | $6.306_{\pm 0.007}$ | $2.103_{\pm 0.005}$ | $\mathbf{1.985}_{\pm 0.004}$ | $1.833_{\pm 0.009}$ | $0.963_{\pm 0.002}$ | $6.791_{\pm 0.006}$ |
| log_semic | $1.956_{\pm 0.006}$ | $6.425_{\pm 0.006}$ | $6.306_{\pm 0.007}$ | $2.345_{\pm 0.005}$ | $1.921_{\pm 0.004}$ | $2.020_{\pm 0.011}$ | $0.841_{\pm 0.002}$ | $6.638_{\pm 0.007}$ |
| circ | $\mathbf{2.268}_{\pm 0.004}$ | $6.515_{\pm 0.005}$ | $0.000_{\pm 0.000}$ | $1.865_{\pm 0.012}$ | $1.710_{\pm 0.003}$ | $1.782_{\pm 0.011}$ | $\mathbf{1.210}_{\pm 0.002}$ | $6.937_{\pm 0.005}$ |
| disc_circ | $2.211_{\pm 0.005}$ | $2.916_{\pm 0.002}$ | $6.306_{\pm 0.007}$ | $1.726_{\pm 0.019}$ | $1.781_{\pm 0.003}$ | $1.321_{\pm 0.014}$ | $1.138_{\pm 0.002}$ | $2.955_{\pm 0.002}$ |
| clust | $0.462_{\pm 0.001}$ | $\mathbf{6.846}_{\pm 0.005}$ | $\mathbf{6.306}_{\pm 0.007}$ | $1.159_{\pm 0.003}$ | $0.584_{\pm 0.002}$ | $1.120_{\pm 0.002}$ | $0.892_{\pm 0.002}$ | $\mathbf{6.979}_{\pm 0.005}$ |
| *Qwen2.5-3B-IT* | | | | | | | | |
| lin | $2.395_{\pm 0.014}$ | $7.343_{\pm 0.011}$ | $7.093_{\pm 0.010}$ | $1.994_{\pm 0.005}$ | $1.715_{\pm 0.004}$ | $1.859_{\pm 0.008}$ | $0.557_{\pm 0.003}$ | $7.816_{\pm 0.009}$ |
| log_lin | $1.903_{\pm 0.013}$ | $7.360_{\pm 0.011}$ | $7.093_{\pm 0.010}$ | $\mathbf{2.719}_{\pm 0.005}$ | $1.632_{\pm 0.004}$ | $\mathbf{3.576}_{\pm 0.006}$ | $0.532_{\pm 0.005}$ | $7.894_{\pm 0.010}$ |
| semic | $2.736_{\pm 0.011}$ | $7.672_{\pm 0.010}$ | $\mathbf{7.093}_{\pm 0.010}$ | $2.097_{\pm 0.007}$ | $\mathbf{1.890}_{\pm 0.004}$ | $1.920_{\pm 0.011}$ | $0.813_{\pm 0.003}$ | $8.194_{\pm 0.009}$ |
| log_semic | $2.577_{\pm 0.012}$ | $7.534_{\pm 0.010}$ | $7.093_{\pm 0.010}$ | $2.295_{\pm 0.008}$ | $1.822_{\pm 0.004}$ | $2.131_{\pm 0.010}$ | $0.713_{\pm 0.003}$ | $8.064_{\pm 0.009}$ |
| circ | $\mathbf{2.860}_{\pm 0.010}$ | $7.793_{\pm 0.010}$ | $0.000_{\pm 0.000}$ | $1.604_{\pm 0.019}$ | $1.578_{\pm 0.003}$ | $1.478_{\pm 0.008}$ | $\mathbf{1.134}_{\pm 0.002}$ | $8.250_{\pm 0.008}$ |
| disc_circ | $2.347_{\pm 0.011}$ | $2.893_{\pm 0.002}$ | $7.093_{\pm 0.010}$ | $1.241_{\pm 0.026}$ | $1.561_{\pm 0.003}$ | $0.725_{\pm 0.015}$ | $0.972_{\pm 0.005}$ | $2.858_{\pm 0.001}$ |
| clust | $0.508_{\pm 0.002}$ | $\mathbf{7.958}_{\pm 0.009}$ | $\mathbf{7.093}_{\pm 0.010}$ | $1.147_{\pm 0.002}$ | $0.620_{\pm 0.002}$ | $1.023_{\pm 0.002}$ | $0.895_{\pm 0.002}$ | $\mathbf{8.416}_{\pm 0.007}$ |
| *gemma-2-2b-IT* | | | | | | | | |
| lin | $2.007_{\pm 0.010}$ | $6.408_{\pm 0.009}$ | $6.726_{\pm 0.010}$ | $1.868_{\pm 0.005}$ | $1.650_{\pm 0.004}$ | $1.945_{\pm 0.010}$ | $0.867_{\pm 0.005}$ | $6.476_{\pm 0.009}$ |
| log_lin | $1.599_{\pm 0.008}$ | $6.249_{\pm 0.009}$ | $6.726_{\pm 0.010}$ | $\mathbf{3.269}_{\pm 0.005}$ | $1.661_{\pm 0.004}$ | $\mathbf{3.655}_{\pm 0.007}$ | $0.616_{\pm 0.006}$ | $6.542_{\pm 0.010}$ |
| semic | $2.376_{\pm 0.011}$ | $6.598_{\pm 0.008}$ | $\mathbf{6.726}_{\pm 0.010}$ | $2.047_{\pm 0.005}$ | $\mathbf{1.871}_{\pm 0.004}$ | $2.020_{\pm 0.014}$ | $1.091_{\pm 0.005}$ | $6.845_{\pm 0.009}$ |
| log_semic | $2.226_{\pm 0.007}$ | $6.468_{\pm 0.008}$ | $6.726_{\pm 0.010}$ | $2.392_{\pm 0.005}$ | $1.819_{\pm 0.004}$ | $2.252_{\pm 0.016}$ | $0.946_{\pm 0.005}$ | $6.703_{\pm 0.010}$ |
| circ | $\mathbf{2.605}_{\pm 0.007}$ | $6.490_{\pm 0.007}$ | $0.000_{\pm 0.000}$ | $1.481_{\pm 0.005}$ | $1.616_{\pm 0.003}$ | $1.782_{\pm 0.027}$ | $\mathbf{1.184}_{\pm 0.003}$ | $6.921_{\pm 0.009}$ |
| disc_circ | $2.078_{\pm 0.008}$ | $2.834_{\pm 0.001}$ | $6.726_{\pm 0.010}$ | $1.281_{\pm 0.006}$ | $1.601_{\pm 0.004}$ | $1.523_{\pm 0.045}$ | $1.023_{\pm 0.004}$ | $2.851_{\pm 0.002}$ |
| clust | $0.589_{\pm 0.002}$ | $\mathbf{6.847}_{\pm 0.006}$ | $6.726_{\pm 0.010}$ | $1.189_{\pm 0.002}$ | $0.768_{\pm 0.003}$ | $1.556_{\pm 0.003}$ | $1.114_{\pm 0.004}$ | $\mathbf{7.036}_{\pm 0.008}$ |
| *Llama-3.2-3B* | | | | | | | | |
| lin | $1.451_{\pm 0.005}$ | $6.035_{\pm 0.008}$ | $6.401_{\pm 0.007}$ | $1.800_{\pm 0.005}$ | $0.521_{\pm 0.011}$ | $1.569_{\pm 0.009}$ | $0.702_{\pm 0.003}$ | $6.164_{\pm 0.009}$ |
| log_lin | $0.952_{\pm 0.005}$ | $5.909_{\pm 0.008}$ | $6.401_{\pm 0.007}$ | $\mathbf{2.717}_{\pm 0.004}$ | $0.409_{\pm 0.016}$ | $\mathbf{3.429}_{\pm 0.006}$ | $0.639_{\pm 0.004}$ | $6.232_{\pm 0.010}$ |
| semic | $1.712_{\pm 0.005}$ | $6.303_{\pm 0.007}$ | $\mathbf{6.401}_{\pm 0.007}$ | $2.090_{\pm 0.005}$ | $0.673_{\pm 0.007}$ | $1.856_{\pm 0.007}$ | $0.971_{\pm 0.003}$ | $6.541_{\pm 0.008}$ |
| log_semic | $1.533_{\pm 0.005}$ | $6.134_{\pm 0.008}$ | $6.401_{\pm 0.007}$ | $2.269_{\pm 0.005}$ | $0.574_{\pm 0.008}$ | $2.041_{\pm 0.008}$ | $0.844_{\pm 0.003}$ | $6.398_{\pm 0.009}$ |
| circ | $\mathbf{1.862}_{\pm 0.004}$ | $6.282_{\pm 0.006}$ | $0.000_{\pm 0.000}$ | $1.882_{\pm 0.006}$ | $\mathbf{0.879}_{\pm 0.010}$ | $1.760_{\pm 0.009}$ | $\mathbf{1.194}_{\pm 0.002}$ | $6.676_{\pm 0.006}$ |
| disc_circ | $1.841_{\pm 0.005}$ | $2.943_{\pm 0.002}$ | $6.401_{\pm 0.007}$ | $1.727_{\pm 0.009}$ | $0.696_{\pm 0.010}$ | $1.291_{\pm 0.012}$ | $1.155_{\pm 0.002}$ | $2.923_{\pm 0.001}$ |
| clust | $0.489_{\pm 0.002}$ | $\mathbf{6.619}_{\pm 0.005}$ | $\mathbf{6.401}_{\pm 0.007}$ | $1.159_{\pm 0.002}$ | $0.804_{\pm 0.007}$ | $1.133_{\pm 0.002}$ | $1.004_{\pm 0.003}$ | $\mathbf{6.787}_{\pm 0.007}$ |
| *Qwen2.5-3B* | | | | | | | | |
| lin | $2.079_{\pm 0.009}$ | $7.424_{\pm 0.008}$ | $6.868_{\pm 0.009}$ | $1.792_{\pm 0.006}$ | $1.460_{\pm 0.008}$ | $1.806_{\pm 0.012}$ | $0.675_{\pm 0.003}$ | $7.835_{\pm 0.009}$ |
| log_lin | $1.579_{\pm 0.009}$ | $7.388_{\pm 0.008}$ | $6.868_{\pm 0.009}$ | $\mathbf{2.823}_{\pm 0.005}$ | $1.449_{\pm 0.008}$ | $\mathbf{3.200}_{\pm 0.006}$ | $0.511_{\pm 0.004}$ | $7.921_{\pm 0.009}$ |
| semic | $2.383_{\pm 0.009}$ | $7.705_{\pm 0.007}$ | $6.868_{\pm 0.009}$ | $2.138_{\pm 0.005}$ | $\mathbf{1.592}_{\pm 0.007}$ | $1.403_{\pm 0.011}$ | $0.946_{\pm 0.003}$ | $8.180_{\pm 0.007}$ |
| log_semic | $2.240_{\pm 0.009}$ | $7.578_{\pm 0.008}$ | $\mathbf{6.868}_{\pm 0.009}$ | $2.318_{\pm 0.005}$ | $1.551_{\pm 0.005}$ | $1.451_{\pm 0.013}$ | $0.805_{\pm 0.003}$ | $8.069_{\pm 0.008}$ |
| circ | $\mathbf{2.522}_{\pm 0.005}$ | $7.682_{\pm 0.007}$ | $0.000_{\pm 0.000}$ | $1.873_{\pm 0.006}$ | $1.340_{\pm 0.004}$ | $0.646_{\pm 0.009}$ | $\mathbf{1.220}_{\pm 0.002}$ | $8.204_{\pm 0.006}$ |
| disc_circ | $2.003_{\pm 0.006}$ | $2.855_{\pm 0.001}$ | $6.868_{\pm 0.009}$ | $1.613_{\pm 0.010}$ | $1.506_{\pm 0.005}$ | $-0.481_{\pm 0.026}$ | $1.028_{\pm 0.003}$ | $2.862_{\pm 0.001}$ |
| clust | $0.518_{\pm 0.002}$ | $\mathbf{7.935}_{\pm 0.007}$ | $6.868_{\pm 0.009}$ | $1.157_{\pm 0.002}$ | $0.642_{\pm 0.003}$ | $1.306_{\pm 0.002}$ | $0.937_{\pm 0.002}$ | $\mathbf{8.371}_{\pm 0.006}$ |
| *gemma-2-2b* | | | | | | | | |
| lin | $2.353_{\pm 0.009}$ | $6.176_{\pm 0.010}$ | $6.449_{\pm 0.011}$ | $1.627_{\pm 0.006}$ | $0.609_{\pm 0.020}$ | $1.601_{\pm 0.009}$ | $0.874_{\pm 0.003}$ | $6.449_{\pm 0.009}$ |
| log_lin | $1.622_{\pm 0.009}$ | $6.064_{\pm 0.009}$ | $6.449_{\pm 0.011}$ | $\mathbf{2.615}_{\pm 0.006}$ | $0.497_{\pm 0.025}$ | $\mathbf{3.462}_{\pm 0.006}$ | $0.726_{\pm 0.006}$ | $6.617_{\pm 0.009}$ |
| semic | $2.741_{\pm 0.009}$ | $6.424_{\pm 0.009}$ | $6.449_{\pm 0.011}$ | $1.856_{\pm 0.005}$ | $0.833_{\pm 0.011}$ | $1.837_{\pm 0.008}$ | $1.172_{\pm 0.003}$ | $6.807_{\pm 0.008}$ |
| log_semic | $2.543_{\pm 0.009}$ | $6.274_{\pm 0.008}$ | $\mathbf{6.449}_{\pm 0.011}$ | $2.090_{\pm 0.006}$ | $0.675_{\pm 0.012}$ | $2.109_{\pm 0.007}$ | $1.032_{\pm 0.003}$ | $6.725_{\pm 0.009}$ |
| circ | $\mathbf{2.810}_{\pm 0.007}$ | $6.649_{\pm 0.007}$ | $0.000_{\pm 0.000}$ | $1.501_{\pm 0.006}$ | $\mathbf{0.842}_{\pm 0.013}$ | $1.319_{\pm 0.009}$ | $\mathbf{1.404}_{\pm 0.004}$ | $6.881_{\pm 0.008}$ |
| disc_circ | $2.194_{\pm 0.007}$ | $2.826_{\pm 0.001}$ | $6.449_{\pm 0.011}$ | $1.287_{\pm 0.009}$ | $0.680_{\pm 0.020}$ | $0.657_{\pm 0.016}$ | $1.153_{\pm 0.005}$ | $2.832_{\pm 0.001}$ |
| clust | $0.518_{\pm 0.002}$ | $\mathbf{6.907}_{\pm 0.007}$ | $6.449_{\pm 0.011}$ | $1.282_{\pm 0.003}$ | $0.796_{\pm 0.010}$ | $1.222_{\pm 0.003}$ | $0.977_{\pm 0.002}$ | $\mathbf{7.024}_{\pm 0.007}$ |

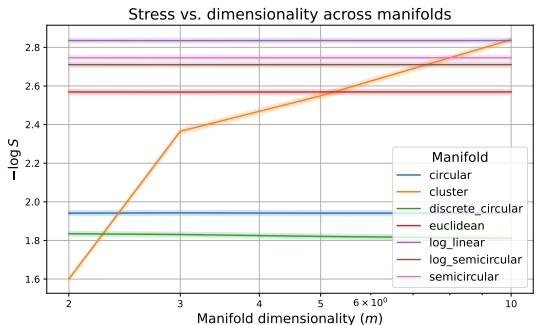

Figure 17: Additional critical difference diagrams showing avg. rank of manifolds across all models over 500 bootstrapping iterations. Horizontal bars show groups of statistical equivalence. Best-ranking manifold is always statistically different from others, the only exception being the `date_temperature` dataset.

Figure 18: Stress as a function of subspace dimensionality $m$, grouped by manifold. The **cluster** manifold displays an upward trend while all others are stable.

Figure 19: Stress as a function of subspace dimensionality $m$, grouped by model. Scores remain largely stable across models and dimensionalities.

### D.4 Sensitivity to Number of Components $m$

We analyze the effect of the recovered subspace dimensionality $m$ on the stress score by varying $m \in \{2, 3, 4, 5, 10\}$ and measuring stress across all tasks and models. Figure 18 shows that stress is largely stable as dimensionality increases for nearly all manifold hypotheses. The main exception is the **cluster** hypothesis, whose stress increases with $m$. This behavior follows from its distance definition, which enforces equal pairwise distances between the $|y|$ clusters: in $m$ dimensions, at most $m + 1$ clusters can be embedded with equal separation, so tasks with more labels necessarily incur higher stress at fixed dimensionality. In contrast, the other manifold hypotheses do not depend on label cardinality and therefore show near-invariant stress across dimensions. This pattern is mirrored across models in Figure 19, where stress remains consistent once the **cluster** case is excluded.

Based on these observations, we fix the SMDS dimensionality to $m = 3$ throughout the study. This is the smallest value sufficient to represent all considered manifold hypotheses: linear and cyclical structures require at most two dimensions, while cluster structures in tasks such as `date_season` and `time_of_day_phase` involve four labels ($|y| = 4$) and thus require three dimensions to be embedded with equal separation. Increasing $m$ beyond this threshold yields only minor changes in stress for all but the **cluster** manifold. We recommend choosing a single shared dimensionality to ensure consistent comparisons across manifold types and model settings.

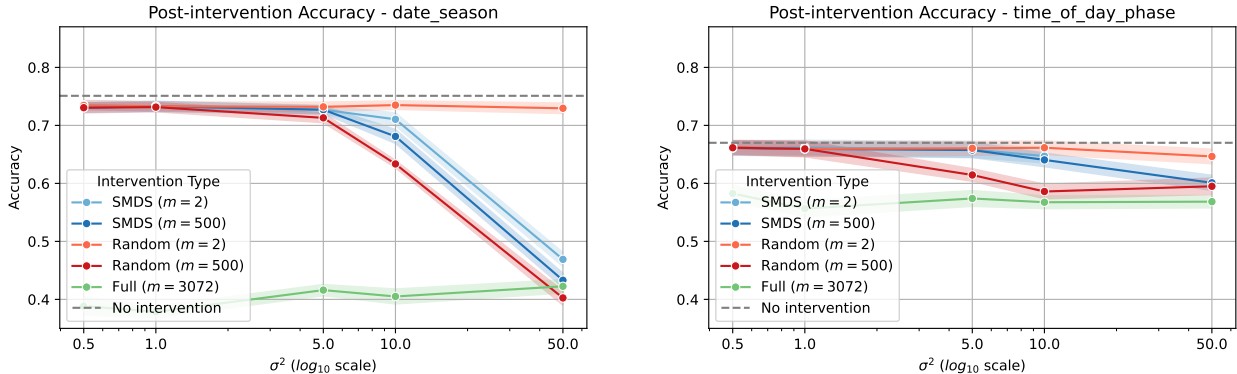

Figure 20: Additional accuracy plots from the intervention experiment. Error bars represent standard error. The `time_of_day` task is the least affected by all forms of intervention.

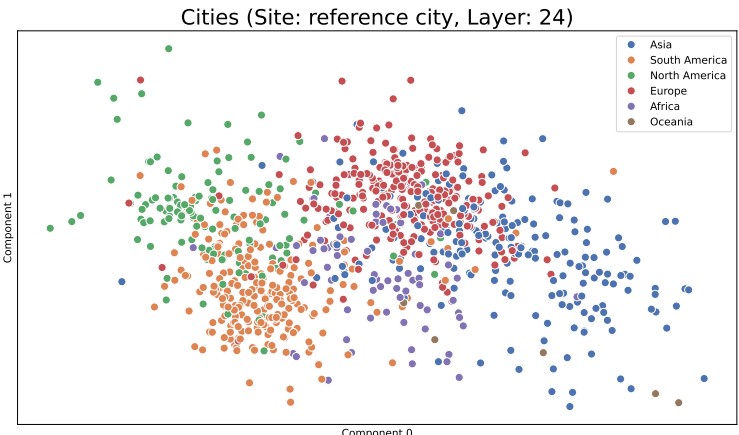

Figure 21: SMDS of gemma-2-2b-it on the `cities` task. The recovered projection shows the relative position of continents. For the sake of clarity, the flat manifold is shown instead of the best-scoring spherical one.

### D.5  Additional Observations on Intervention

Figure 20 shows how the `time_of_day` is the least affected by intervention, even when perturbing the full latent space. We believe this is due to the specific formatting of time used: expressions such as `19:37` are tokenized as `19`, `:`, `37`, with the TE site corresponding to the minute part of the expression. For most examples, the hour is sufficient to determine the right answer, and since that information is left untouched, the model is able to continue with minimal disruption.

### D.6  Identifying a Spatial Manifold

To show the flexibility of SMDS, we extend our analysis to a spatial reasoning task. In the same vein as Appendix B, we build sentences composed of three statements "`<name>` lives in `<city>`." Then, we prepend a continuation "The person who lives closest to `<name>` is" to elicit reasoning. Names are sampled from the usual set, while cities are obtained from the World Cities Database[5]. We select only prominent cities as they are more likely to be present in the model's memory. For the US and Canada we instead select cities with > 10.000 inhabitants, since following the provided labels results in severe undersampling. We then uniformly sample cities based on location.

---

[5]https://simplemaps.com/data/world-cities

Each city is characterized by its latitude and longitude coordinates $c_i = (\text{lat}_i, \text{lon}_i)$. From these, we project cities on various shapes and compute the relative distance function. We investigate a flat plane, a sphere, a cylinder, and a complex geometry defined by the geodesic distance between cities. The flat manifold is computed simply as the Euclidean distance between the two coordinates, same as the `linear` metric used before. For the sphere manifold, we convert each coordinate into a 3D point on a sphere of radius $r$ as follows:

$$\phi_i = \text{radians}(\text{lat}_i), \quad \lambda_i = \text{radians}(\text{lon}_i),$$
$$x_i = r\cos(\phi_i)\cos(\lambda_i),$$
$$y_i = r\cos(\phi_i)\sin(\lambda_i),$$
$$z_i = r\sin(\phi_i).$$

Then the distance between two cities is the Euclidean chord length:

$$\|\delta_{ij}\| = \sqrt{(x_i - x_j)^2 + (y_i - y_j)^2 + (z_i - z_j)^2}.$$

For the cylinder manifold, we map latitude to vertical height and longitude to angle around a cylinder of radius $r$. Each point is embedded as:

$$h_i = \text{radians}(\text{lat}_i) \cdot s, \quad \lambda_i = \text{radians}(\text{lon}_i),$$
$$x_i = r\cos(\lambda_i),$$
$$y_i = r\sin(\lambda_i),$$
$$z_i = h_i.$$

The chord distance is again computed as the Euclidean distance in 3D. For the geodesic manifold, we compute the great-circle distance between two cities—i.e., the shortest path along the surface of a sphere. We first convert latitude and longitude to radians and compute the differences:

$$\phi_i = \text{radians}(\text{lat}_i), \quad \lambda_i = \text{radians}(\text{lon}_i),$$
$$\Delta\phi = \phi_i - \phi_j, \quad \Delta\lambda = \lambda_i - \lambda_j.$$

We then use the Haversine formula:

$$a = \sin^2\left(\frac{\Delta\phi}{2}\right) + \cos(\phi_i)\cos(\phi_j)\sin^2\left(\frac{\Delta\lambda}{2}\right).$$
$$\|\delta_{ij}\| = r \cdot 2\arcsin\left(\sqrt{a}\right).$$

This corresponds to the true surface distance between two points on the Earth, assuming a perfect sphere. In addition to the usual LP and A sites, we analyze two more locations: the correct city (CC), corresponding to the final token of the city where the correct person lives, and the reference city (RC), referring to the final token of the city where the person in the question lives. Both cities are drawn from the context statements.

Table 5 shows that the manifold achieving the closest fit is a spherical one across all models. In Figure 21, we visualize the projection recovered by SMDS and find clear clusters around the shapes of continents. Their relative position is consistent with their real-life location, but projecting a spherical manifold onto a plane inevitably distorts their real position.

### D.7 Feature Manifolds are not Artefacts of SMDS

In this section we validate the robustness of SMDS and confirm that the feature manifolds recovered are indeed consistent and not an artefact of overfitting a projection. Primary evidence is provided in the main analysis of §5: the error bars produced by cross-validation are narrow for almost all datasets, confirming that activations for a given feature do have a preferential manifold.

The second piece of evidence is obtained by designing control tasks following Hewitt & Liang (2019). We build control variants for all tasks by shuffling the labels. This should make it impossible for SMDS to identify a structure and we should observe a significant increase in stress. For each model-task pair, we evaluate the best manifold identified in §5. As in the main experiment, we perform a 5-fold cross-validation on the dataset. Table 10 shows the results: absence of structure causes a sharp increase in stress (and corresponding drop in $-\log S$). This is evidence that SMDS does not force a structure when no underlying manifold exists.

Table 10: Stress values for control tasks. Absolute difference with the base task is shown in red. SMDS consistently produces low scores if no structure is present.

| Dataset | Llama-3.2-3B-Instruct | | Qwen2.5-3B-Instruct | | gemma-2-2b-it | |
|---|---|---|---|---|---|---|
| | Best shape | $-\log S$ | Best shape | $-\log S$ | Best shape | $-\log S$ |
| date | circular | $0.995_{-0.932}$ | circular | $0.855_{-2.075}$ | circular | $0.806_{-1.563}$ |
| date_season | cluster | $0.998_{-4.509}$ | cluster | $0.956_{-7.205}$ | cluster | $0.858_{-6.308}$ |
| date_temperature | cluster | $0.376_{-5.669}$ | cluster | $0.332_{-7.010}$ | cluster | $0.211_{-6.111}$ |
| duration | log_linear | $0.513_{-2.490}$ | log_linear | $0.519_{-2.083}$ | log_linear | $0.445_{-2.059}$ |
| notable | semicircular | $0.722_{-1.248}$ | semicircular | $0.740_{-0.801}$ | semicircular | $0.482_{-1.168}$ |
| periodic | log_linear | $0.523_{-2.888}$ | log_linear | $0.502_{-3.150}$ | log_linear | $0.235_{-3.470}$ |
| time_of_day | circular | $0.987_{-0.223}$ | circular | $0.978_{-0.066}$ | semicircular | $0.683_{-0.468}$ |
| time_of_day_phase | cluster | $0.961_{-4.710}$ | cluster | $1.000_{-6.869}$ | cluster | $0.902_{-6.487}$ |

## D.8 Exploring the Impact of Instruction Tuning

Since we exclusively use instruction-tuned models in our main experiments, we are interested in whether instruction tuning impacts the feature manifolds and accuracies of our models. Instruction tuning is a post-training method that is widely believed to enhance models' generalization and task-solving capabilities (Wei et al., 2021; Chung et al., 2024), however not all of the potential changes induced in a base model by instruction tuning have been explored. Various prior works suggest that instruction tuning mainly impacts stylistic output tokens rather than changing the model's parametric knowledge (Zhou et al., 2023; Ghosh et al., 2024; Lin et al., 2023), and that it additionally causes models to rotate the basis of their representation space to adapt to user-oriented tasks (Wu et al., 2024). However, these works deal primarily with token probability distributions and do not explore feature manifolds.

To explore the impact of instruction tuning in our experimental setup, we conduct our main experiments on the base versions of three of our models: Llama-3.2-3B, Qwen2.5-3B, and Gemma-2-2B. We report the stress values in Table 3 and the feature manifolds in Figure 15.

Overall, across our three models, we find that instruction tuning did not substantially alter the optimal structure. For all tasks except notable, periodic, and duration, the structure that was optimal for the instruction-tuned model tended to remain in the top-3 optimal structures for the base model as well. We note that the three outlier tasks have monotonic topologies, while the rest have cyclical or cluster-like structures.

For some tasks, we observed that the feature manifolds for base models tended to be more scattered than those of the instruction-tuned models. For example, the date manifolds for all instruction-tuned models (Figures 3, 13, 14) have tighter, well-formed ring structures than the manifolds of the base models (Figure 15). However, this was not the case for all tasks: for example, the periodic manifolds for both base and instruction-tuned models showed a clear separation of clusters that remained consistent within a particular model architecture.

Surprisingly, we found that the accuracies varied unpredictably between the base and instruction-tuned models. For Llama, all base accuracies were much lower than the instruction-tuned accuracies, while with the Qwen and Gemma models, base models sometimes markedly outperformed the instruction-tuned models. This could possibly be due to differences in the instruction-tuning methods of these models.

Overall, our results suggest that the impact of instruction-tuning on feature manifolds will depend on the task, model architecture, as well as on the specifics of the instruction-tuning process. A detailed exploration of this is a promising avenue for future work.

