# OpenReview forum: "Hypothesis-Driven Feature Manifold Analysis in LLMs via Supervised Multi-Dimensional Scaling"
_TMLR — Accepted by TMLR_

### Review · Reviewer_eyL6 · 2025-11-04

**Summary Of Contributions:**

In this work, the authors propose a targeted variant of multidimensional scaling (MDS) to discover submanifolds with specific geometric and topological properties consistent with input data, which they call supervised MDS (SMDS). The authors outline specific target properties encoded by various choices of distance functions to test for circular, linear, and categorical concepts to see if these can be identified and correlated to practically analogous concepts in the data. Basing their study on findings in previous works, the authors focus almost exclusively on demonstrating time-based concepts, in addition to a longitude-latitude concept, across what amounts to three model families (Llama, Qwen, Gemma) with some variety in capacity (2-3B, 8B, 70B parameters).

### Note on Structural Concern

Before addressing the scientific claims, this reviewer notes a structural concern: while the paper is formatted as a <12 page short work, the core quantitative evidence supporting the main findings is located in the appendix (Tables 5, 8; Figures 10, 11, 12). Hence the supplementary ceases to act as additional results and instead contains all the primary evidence behind the work. The main text relies predominantly on visual/exploratory presentations that cannot be properly evaluated without cross-referencing extensive appendix content.

For TMLR, which offers flexible paper length, this reviewer questions whether artificially constraining the paper to 12 pages is appropriate when it prevents integrated presentation of the core results. This is an editorial choice to provide for the need for fast review cycles for conference-length papers. In this case, the reviewer believe the work is presented as a shorter work to adhere to the page limit for shorter review cycles. This exaggerates the load of the review, and makes it more challenging to assess whether the quantitative evidence is sufficient to support the claims.

By choosing this format, the authors implicitly signal that the appendix material is supplementary rather than central to evaluating the claims. This review will therefore weight the main text evidence more heavily in assessing whether the findings are adequately supported.


### Findings

The authors present three claimed findings;

1. Temporal concepts / "entities" share an intrinsic manifold structure across different models. The temporal understanding of different models hence seem to mirror across independently trained models. The authors provide quantitative evidence in Figures 3, 10, 11 and Tables 5, 8. The same geometry consistently ranks 1st across models for most tasks (e.g., circular for dates in all 3 models, cluster for date_season in all 3 models). While this seems to outline a general pattern, the evidence lacks statistical significance tests to show that the 1st-ranked geometry is significantly better than the 2nd-ranked. There is no reporting on difference in scores (effect size) nor confidence intervals. While some scores are provided in figures, without quantification of the margins between geometries and their statistical reliability, it's difficult to assess whether the preference for specific geometries is strong or marginal.
2. Language models adapt structure "in-context" for different tasks, such that circular properties are adapted in temporal concepts where appropriate, such as the circular structure of months, and linear structure is applied with perceived temperature (hot / cold). The authors provide evidence through Figure 5 (visualization), Table 5 (showing different best geometries for date vs. date_season vs. date_temperature), and Figure 6 (showing information flow across layers). Overall, the trend seems to be that tasks with the same context but different continuations prefer different geometries at the LP site (circular for date, cluster for seasonal tasks). This holds consistently for Llama and Qwen, though Gemma shows a different pattern for date_temperature. However, the evidence for in-context adaptation (that the model actively transforms one representation into another based on the prompt) is primarily illustrative rather than quantitative. While Figure 5 shows different shapes at TE vs. LP sites, there is no measurement of how different these manifolds are or whether the transformation is statistically significant. Moreover, the evidence doesn't distinguish whether models actively reshape a single representation versus maintaining separate task-specific representations throughout processing.
3. The third finding is that feature manifolds actively play a role in reasoning for language models. The authors provide two pieces of evidence: (a) perturbations to manifold subspaces degrade reasoning performance more than random perturbations (Figure 7); (b) manifold quality (measured by stress) correlates with task accuracy (Spearman's ρ = 0.513, p = 0.0174). Of the three findings, this provides the strongest quantitative evidence in the main text, including the only reported p-values for any of the main claims. The correlation between manifold quality and accuracy provides statistical support for the functional relevance of discovered manifolds. However, the causal intervention experiment (Figure 7), while showing clear degradation patterns, lacks error bars or statistical tests comparing the SMDS perturbation condition against random perturbations. Without significance tests, it's unclear whether the difference between perturbing SMDS-identified subspaces versus random subspaces is reliably greater than noise. Additionally, the intervention is only demonstrated on three task-model pairs selected for high baseline accuracy, severely limiting the generalisability of this finding.

Overall, the findings the authors present are not wholly unconvincing, but the lack of statistical testing and rigour severely lessens the evidence for the claims made in the paper. The fact that the central quantitative results are relegated to the supplementary material also affects the way the manuscript is interpreted and parsed by the reader. In this case, the paper would have benefited from a longer presentation to fully back up the weight of the claims made by the authors.

**Additional Comments:**

To reiterate on the structural concern, we highlight that the paper is currently 11 pages, with a 12 page supplementary material section. The issue the reviewer has with this is that the central quantitative results made to support the claims in the paper is almost exclusively placed in the supplementary section, which makes it difficult for readers to engage with the work as intended.

While this reviewer expects that this is likely due to a wish for a shorter review cycle, it is not a practice that should be encouraged. TMLR supports works that are longer than standard conference papers, but with the additional caveat that reviewers and ACs are provided enough time to meaningfully engage with the work.

**Audience:**

Yes

**Audience Explanation:**

Overall, the reviewer finds the scope of the research in the manuscript both interesting and novel. The paper presents a valuable new approach through SMDS, and observes highly interesting patterns while proposing novel findings that corroborates with existing works. Generally, these types of studies have the potential of providing a lot of value to the research community, and this reviewer does not doubt that if methodological an presentation issues are sufficiently addressed, the work has potential to serve as an interesting and impactful read that would interest a significant majority of the journals intended audience.

To improve on the study, the authors are encouraged to determine some concepts independently of previously discovered concepts to investigate whether the discovered submanifold structures agree with intuition. While this is not required, it would greatly add to the existing study, further demonstrating the applicability of the method to discover new structures instead of just confirming existing works.

**Broader Impact Concerns:**

As a largely theoretical work on machine learning, the ethical implications of the work does not warrant a broader impact statement.

**Claims And Evidence:**

No

**Claims Explanation:**

The authors rely heavily on qualitative visualisations and exploratory data analysis to make their claims, as supposed to more rigorous quantitative hypothesis testing. Given the strong claims, the findings need to be backed up with more rigour than visualisation to make the claims. If these claims hold, there should be a clear way to do this testing and provide statistical test results to verify the claims made beyond visual inspection. Moreover, given that the authors have opted to present the main quantitative results as supplementary to keep the main page count low, the evidence for the claims made in the paper is in this reviewers opinion therefore insufficient.

**In summary**: the claims are supported by suggestive patterns but lack the statistical rigor required for convincing scientific evidence. The work would benefit from formal hypothesis testing throughout, multiple testing correction, effect size reporting, and either integration of appendix results into the main narrative or reframing of claims to match the level of evidence provided.

### Issues with Claimed Findings

For F1; the qualitative pattern seems clear, as the same geometries tend to rank first across models. However, without hypothesis tests, effect sizes, or confidence intervals, we cannot determine if these preferences are statistically meaningful or within measurement noise.

In the case of F2, the evidence is primarily illustrative. While visualizations show different geometries at different processing stages, there is no quantification of manifold transformations or statistical tests showing these changes are systematic rather than artifacts of visualization.

F3 represents the strongest claim, with correlation statistics (p = 0.0174) and intervention experiments. However, even here, the intervention lacks error bars and significance tests comparing conditions, and is demonstrated on only three carefully selected task-model pairs.

### Issue on Temporal Focus

The paper focuses almost exclusively on temporal reasoning because *"initial evidence has found temporal feature manifolds to vary widely across tasks"* (page 2), citing Engels et al. (2025) and Heinzerling & Inui (2024). While this initially seemed like a strong motivation, it also creates potential circularity:
- prior work found circular structures for temporal concepts,
- SMDS is designed to detect circular structures,
- the paper tests primarily on temporal tasks,
- circular structures are found.

In other words, the authors are implicitly looking to confirm rather than test for existing findings relating to submanifold structures. This further highlights the need for explicitly defining the hypotheses of the work, and ensuring the reader that the scientific methodology of the paper is sound and unbiased.

While the authors add an additional test on spatiotemporal structure, this too has been largely confirmed in previous works. To strenghten the study, additional concepts with specific structures could be tested to verify these structures to confirm that the structures appear outside of previously discovered concepts. This would improve the novelty and contribution of the work.

**Requested Changes:**

### Main concerns
- Add formal statistical hypothesis testing for all three main findings, and clearly outline the framework and hypotheses in the study. This is required for recommending acceptance.
- Integrate the central quantitative results into the main body of the paper. If the statistical tests are performed, and corroborate the findings in the work, these should be sufficient to include in the main body, and the current details can remain in the supplementary material as additional evidence. However, without these tests, the central evidence for the claims should not be placed in the supplementary. This is required for this reviewer to recommend acceptance.
- Quantify cross-model consistency with explicit metrics (e.g., rank correlation of geometry preferences, top-k agreement rates across models) rather than relying on visual inspection. Given the nature of the claims as highly general findings relating to the geometry of concepts in language models, showing this explicitly is required to back up the claims made in the paper.

### Optional Addendums
- It would be interesting to conduct additional experiments for at least one circular and log linear concept, independent of previous works. One example that springs to mind is testing for colour hue, which humans interpret as circular. An example of a log-linear magnitude type concept is easier to determine, e.g. physical size (animals, common objects, physical structures or buildings).

---

> ### Author Response · Authors · 2025-11-28
>
> We thank the reviewer for the detailed feedback on structure and content. In this comment we detail the significant changes we made to accommodate the reviewer’s suggestion.
>
> > Add formal statistical hypothesis testing for all three main findings + Quantify cross-model consistency with explicit metrics (e.g., rank correlation of geometry preferences, top-k agreement rates across models)
>
> As requested we have added statistical testing for Sections 5.1, 5.4, and reworked Figure 7 to show confidence intervals.
>
> > Integrate the central quantitative results into the main body of the paper.
>
> We have made structural changes by updating the stress table and moving it into the main text. Similarly, Figures and tables from F4 have been moved to main text. To make space and fit into the 12 page limit of this format we moved Related Works to appendix, as the reviewer points out experimental evidence should be prioritized in the main text. We truly appreciate the reviewer's comments. These changes have substantially increased the presentation and clarity of the paper.
>
> > Conduct additional experiments for at least one circular and log linear concept, independent of previous works (color hue, physical size)
>
> We appreciate the proposed extended experimental plan. However, designing and running a new dataset (with statistical significance analysis) is beyond the scope of the current paper, and it is not realistic to achieve such results within the response period. Beyond this, our paper focused on temporal expressions: in Section 5.4 we do present results from other domains to further reinforce that SMDS makes sense outside of the time domain. But this is purely a sanity check. In future we will do more extensive benchmarking across domains.

---

### Review · Reviewer_q9Y9 · 2025-11-10

**Summary Of Contributions:**

This paper proposes a novel interpretability method named SMDS (Supervised Multi-Dimensional Scaling) for automatically discovering feature manifolds in the activation space of Large Language Models (LLMs). SMDS frames manifold discovery as a supervised learning problem, learning a linear projection that maps high-dimensional activations to a low-dimensional space to match a pre-assumed geometric distance (e.g., "circular" or "linear").

Strengths:

Quantitative Comparison:
The main contribution of this paper is providing a unified framework and a quantitative metric (Stress value) that allows researchers to compare the goodness-of-fit of different geometric assumptions. This differs from previous methods that rely on single, fixed assumptions (e.g., PCA, LDA).

Causal Evidence ($\mathcal{F}_{3}$):
The paper’s most compelling finding is in Section 5.3. Through causal intervention (injecting noise into the specific low-dimensional subspace found by SMDS), the authors demonstrate that the LLM relies on these minimal subspaces for reasoning, whereas perturbing random subspaces has little or no effect.

Dynamic Analysis ($\mathcal{F}_{2}$):
The paper presents suggestive evidence that manifold structures may be dynamically adjusted depending on the task (e.g., date vs. date_season).

Weaknesses:

Limited External Validity:
The study depends heavily on synthetic datasets with highly uniform syntactic structures. This limitation makes it unclear whether the discovered manifolds reflect abstract temporal concepts or merely overfitting to specific syntactic templates.

Methodological Simplification:
SMDS functions more as a “hypothesis-driven” linear probe—it tests pre-defined manifold assumptions rather than discovering them automatically. It also struggles with less clearly defined temporal concepts (such as “morning”), as acknowledged in the limitations section.

Data Design Effects:
Some findings, such as “logarithmic compression,” may partially arise from the experimental data design rather than from the model’s internal representations. For example, the label categories (“daily,” “weekly,” “yearly”) are themselves logarithmically spaced.

**Additional Comments:**

This paper addresses an important and timely problem in LLM interpretability.
The strongest component is the causal intervention experiment ($\mathcal{F}_{3}$), which compellingly shows that information may be represented in very low-dimensional subspaces within the LLM—a contribution of genuine independent value.

However, the reliance on synthetic data with fixed syntax substantially limits the generalizability of the findings.
A promising future direction would be to integrate techniques from Topological Data Analysis (e.g., persistent homology) to discover manifold structures in an unsupervised and assumption-free manner, rather than verifying pre-defined shapes such as 2D circles.

**Audience:**

Yes

**Audience Explanation:**

Yes. Mechanistic interpretability remains a central topic in machine learning research.
The paper’s question—how LLMs represent abstract concepts internally—is of strong interest to the TMLR audience.

The causal intervention approach ($\mathcal{F}_{3}$) is technically rigorous and could inspire future work.
Despite certain methodological limitations, the SMDS framework as a “quantitative probe” and the phenomena it highlights are likely to spark useful discussion and follow-up research.

**Broader Impact Concerns:**

The experimental design implicitly reflects a Northern Hemisphere bias (e.g., associating December with “cold”), as noted in the Limitations section.

It would be valuable to expand this discussion in the main text and to emphasize that SMDS could serve as a diagnostic tool to analyze how models encode such cultural or environmental priors.

**Claims And Evidence:**

No

**Claims Explanation:**

Several of the paper’s key claims appear to rely on experimental designs that may introduce confounding factors, leading to conclusions that are less convincing than intended.

1. Limited Generalization Testing

Fixed Syntactic Templates:
All experiments are based on synthetic data with very similar formats (e.g., [Name] [Action] on [Date]).

Concept vs. Template:
The claim that models “converge to similar structures” ($\mathcal{F}_{1}$) does not yet demonstrate convergence to general conceptual representations. Instead, it may reflect that when all models are tested on nearly identical templates, they naturally converge to similar internal shortcuts.

The paper does not yet provide evidence showing whether these manifolds persist under varied syntactic structures (e.g., “The day ... was January 16th”).

2. Methodological Simplification and Over-interpretation of Results

Treating “seasons” as independent clusters may represent a simplification that overlooks their intrinsic cyclical nature. The date_season task currently treats seasons as uncorrelated clusters, ignoring the cyclic order (Spring → Summer → Autumn → Winter). While this design simplifies the experiment, it may also weaken the correspondence between task structure and the intended goal of identifying intuitive and meaningful properties.

Insufficient Evidence for 2D Manifold (Fig. 14):
Section 5.4 claims the discovery of a 2D manifold for “start date” and “duration.”

Methodological Clarification:
The paper would benefit from a clearer explanation of how the “cyclical” and “log-linear” features were combined into a single distance function.

The evidence in Figure 14 appears to challenge the stated interpretation: the relationship between Component 0 and “duration” seems closer to inverse than proportional, with considerable variance in the data points (e.g., “3 days” appears left of “7 days”).

Conflating Memorization and Reasoning:
The “spherical manifold” observed in the cities task may primarily reflect retrieval of memorized geographic information rather than abstract reasoning.

3. Narrative Inconsistency in $\mathcal{F}_{2}$ Findings

The $\mathcal{F}_{2}$ finding contains an internal inconsistency.
On one hand, Figure 5 suggests manifolds are “reshaped” (e.g., “circle” becomes “clusters”).
On the other hand, Figure 6 suggests they are “preserved and propagated” (e.g., “circle” becomes a “blurred circle”).
The discussion appears to merge two distinct mechanisms—“transformation” and “propagation”—that might be analyzed more clearly as separate phenomena.

4. Contextual Interference in TE Activations

The analysis relies on extracting activations from the TE (Temporal Expression) position to study “information flow.” However, in Transformer architectures, the activation at the TE position inherently integrates contextual information and may not represent purely temporal content. This contextual mixing makes the interpretation of “manifolds being preserved and propagated” ambiguous.
The “blurriness” of the manifolds in Figure 6 (higher Stress) could arise from stronger contextual interference during later-stage aggregation. The authors could discuss this issue more explicitly.

**Requested Changes:**

Critical:

Syntax Generalization Test:
To strengthen the main claim, the authors are encouraged to test whether the identified manifolds remain stable when using sentences with different syntactic structures but the same temporal meaning. This would directly assess generalization beyond the fixed templates.

Re-evaluate the “Season” Manifold:
Re-analyze the date_season task using a more appropriate distance function (e.g., discrete_circular) to reflect the cyclical order of seasons, rather than treating them as unordered clusters.

Clarify the 2D Manifold Methodology:
Provide a clearer explanation in Section 5.4 of how the “cyclical” and “log-linear” dimensions were combined into a single 2D distance function. Without this clarification, the results in Figure 14 remain difficult to verify.

Strengthen:

Clarify the $\mathcal{F}_{2}$ Narrative:
Resolve the apparent contradiction in Section 5.2—does the model “propagate” or “reshape” the manifold? Treating these mechanisms separately could improve clarity.

Tone Down Claims:

The finding from the cities task might partly reflect knowledge retrieval or memorization rather than genuine reasoning. Testing with fictional or inconsistent city names could help clarify this.

It would also be useful to note explicitly in the main text (not only in the limitations) that SMDS currently cannot capture fuzzy or context-dependent temporal concepts, which limits its scope.

The considerable variance among data points in Figure 14 should be discussed, even if it arises from contextual ambiguity.

Finally, the authors should acknowledge that the labels used are themselves logarithmically spaced, meaning the observed “logarithmic compression” effect is likely an artifact of the experimental setup rather than an emergent model behavior.

---

> ### Author Response · Authors · 2025-11-28
>
> We thank the reviewer for recognizing the value of our quantitative framework, and for considering Findings 2-3 compelling and motivated by our experiments. We will now clear some misunderstandings and motivate our design choices.
>
> > All experiments are based on synthetic data with very similar formats
>
> The templates we used may not cover all real-world scenarios, however, we took several extra steps to ensure our datasets were as varied as possible. This included randomizing names, actions, and numerical data in the prompt. Our datasets also target different types of reasoning, like spans, direct comparisons, cyclical reasoning, memory recall, as well as an entirely different domain. In all these cases, we observe our findings still hold. Additionally, we have added in Appendix B a comparison of other datasets used in similar peer-reviewed studies on manifold discovery and find our data exhibits greater variability.
>
> >Concept vs. Template
>
> We respectfully disagree with the reviewer's statement that models do not "converge to similar structure," but "converge to similar internal shortcuts for nearly identical templates." Instead, our experiments ran on models with different scales, from different vendors, that therefore implied different optimisation procedures, training corpora, etc. We find it unlikely to see such regularity across all models only because the templates are similar. We further reinforce this by pointing out that templates differ across all eight datasets, and even across setups, results are consistent. We believe this is enough evidence to claim that models have similar internal representationst.
>
> > Re-evaluate the “Season” Manifold
>
> The manifold was indeed evaluated under the discrete_circular manifold, as can be seen from Figure 3. According to our experimental data, the discrete_circular hypothesis simply ranks lower than the alternatives and therefore a different one was used in the analysis.
>
> >  Clarify the 2D Manifold Methodology
>
> We thank the reviewer for pointing out the lack of clarity. We did not experiment with cyclical and log-linear functions: we considered two labels, interpreted as 2D vectors, and used the linear distance function, the only one supporting vector labels. We clarified this in Section 5.4.
>
> > Clarify the F2 Narrative
>
> We apologize for the confusion. We believe the two mechanisms (copying of manifolds to later layers and reshaping) to be related: depending on task and presented information, models reshape and adapt internal representations across tokens. We have made changes in Section 5.2 to make this clearer.
>
> > Contextual Interference in TE Activations
>
> We thank the reviewer for the observation. Indeed, contextual information may affect manifolds by making them noisier. We added a discussion in Section 5.2.
>
> > finding from the cities task might partly reflect knowledge retrieval or memorization rather than genuine reasoning
>
> For both the cities and the notable tasks, this was precisely the intent: provide the model with a diverse set of tasks that probe knowledge retrieval as well as pure reasoning. We argue this is not a downside of our setup but an interesting observation that models build manifolds for both pure reasoning and mnemonic one.
>
> > SMDS currently cannot capture fuzzy or context-dependent temporal concepts
>
> We thank the reviewer for highlighting a limitation shared by all existing supervised dimensionality reduction algorithms, including ours: their inability to capture manifolds associated with "fuzzy" concepts. There is no straightforward way to treat a point with no label, and, to the best of our knowledge, no prior work on the topic either. We point out an interesting fact: in the “time_of_day_phase”, “date_season”, and “date_temperature” datasets, our queries to the models are more ambiguous. As a result, SMDS returned the least specific manifold we had, “cluster.” We can interpret this as the lowest-resolution manifold, but since fuzzy concepts are not part of the paper, we cannot claim this with certainty.
>
> > The considerable variance among data points in Figure 14 should be discussed
>
> The reviewer’s comment stems from our lack of evidence, let’s clarify further: while it is true that some selected points lay further from their expected location in the manifold, one must consider the recovered structure as a whole. In this case, we evaluate manifold quality and find scores significantly better than their control counterparts. We nonetheless have added a new table and reworked the figure in Section 5.4, where we now better clarify employed distance function, stress scores, and the results of a Wilcoxon test showing results are statistically significant across models.
>
> > observed “logarithmic compression” effect is likely an artifact of the experimental setup rather than an emergent model behavior
>
> We thank the reviewer for pointing this out, we have updated our claims in Section 5.1 to acknowledge this.

---

### Review · Reviewer_RQND · 2025-11-14

**Summary Of Contributions:**

The paper uses an extension of Multi-Dimensional Scaling (MDS) called Supervised MDS, to automatically discover the best manifold representation with specific geometric properties for a language modeling task of temporal question answering. The experiments show that across different model sizes and architectures (tested with Qwen, Gemma, Llama family of models), these representations have consistent structures. The experiment also finds that language models utilize different structures when tasked with different types of reasoning. Perturbation analysis on these feature manifolds further show that the language models indeed utilize these feature manifolds for reasoning tasks.

Strengths:
- Clear set of contributions and well-organized content
- Detailed description of experimental setups
- Supporting experimental results

Weaknesses:
- In Figure 5, the standard in which the visualized layers are selected among all other layers is not clear. If they are cherry-picked, further justification is needed.
- In Section 5.3, the justification behind the injection of noise into the first layer is lacking.

**Audience:**

Yes

**Audience Explanation:**

Individuals interested in mechanistic interpretability and geometric structures of manifolds learned by LLMs may be of interest. Temporal reasoning is the task of interest in this paper, so researchers interested in extending to more general reasoning tasks may also be interested in this direction. Also compared to more “famous” methods like SAEs, this method is much more compute efficient, so interpretability researchers with stricter compute budgets may be interested.

**Broader Impact Concerns:**

No impact or ethical concerns.

**Claims And Evidence:**

Yes

**Claims Explanation:**

All main claims are mostly supported by the experimental results.

- The first claim (F1) is supported by the experiments on Gemma, Llama, Qwen family of models in Figure 3 and section 5.1.
- F2 is supported by Figure 5 mainly (although a bit more details on the layers visualized may be needed).
- F3 is supported by the perturbation analysis in Section 5.3 (although further justification of the noise choice may be needed).

Points two and three are further mentioned in Requested Changes; other than these the claims are supported by the experimental results provided.

**Requested Changes:**

- Critical: as mentioned in the Weaknesses part, more justification behind the specific choice of layers visualized in Figure 5 is needed. Are they selected based on the stress value, like done in Figure 3? What do the manifolds in other layers actually look like?
- Critical: as mentioned in the Weaknesses part, more justification behind the choice of noise injection layer as the first layer is needed. Why not inject it to the layer that has the best stress value?
- Recommendation: The concrete results of section 5.4 are all essentially presented in the Appendix. For the paper to be more self-contained, a suggested edition would be to either (1) move the entire section to the appendix and briefly mention the extension to multi-dimensional case in the main part of the paper, or (2) include all figures in the main body of the section so that the results and figures can be better referred to with the main text.
- Recommendation: I think the train/test split details for F1 and F2 is missing. How many data points were held out to compute the stress?

---

> ### Author Response · Authors · 2025-11-28
>
> We thank the reviewer for acknowledging our content was well organised, and our presented evidence detailed and thorough. We would now like to address the comments and clear any misunderstandings.
>
> > More justification behind the specific choice of layers visualized in Figure 5 is needed
>
> Figure 6, as done in Figure 3, shows the layer with the best manifold (i.e. where stress is lowest). For instances of manifolds at other layers please check the top row of Figure 9, which shows instances of manifolds at different depths through the network. We added a more informative description at the end of Section 4.
>
> > More justification behind the choice of noise injection layer as the first layer is needed
>
> We chose Layer 1 as it was the earliest our SMDS could detect manifolds and therefore inform an intervention. We experimented briefly with the layer scoring the best, but did not observe as strong an effect. The reason for this we believe is that information propagates quickly across tokens and layers, therefore the model is able to reconstruct a manifold from context tokens even if its source is missing. We limited ourselves to single layer, single token interventions but probably extending to more context tokens would yield a stronger effect.
>
> > concrete results of section 5.4 are all essentially presented in the Appendix
>
> This is a great suggestion! We have moved the relevant figures and tables into the main text, and we believe this substantially improves the clarity of our presentation.
>
> > train/test split details for F1 and F2 is missing
>
> As mentioned in Section 4, we compute all scores in a K-Fold fashion with K=5. This means that, given a set of N activations X, we partition it in 5 splits, alternately train SMDS on 4 splits and compute scores on the remaining fifth, and finally average the five scores together to get a robust estimate of stress for unseen samples. In classical ML terms, manifold discovery is equivalent to the hyperparameter tuning phase. We have made this clearer in Section 4.

---

> > ### Comment · Reviewer_RQND · 2025-12-12
> > **Best scoring layer != Most influential layer to intervene ?**
> >
> > Thank you for the revision.
> >
> > > We chose Layer 1 as it was the earliest our SMDS could detect manifolds and therefore inform an intervention. We experimented briefly with the layer scoring the best, but did not observe as strong an effect.
> >
> > This is a bit counterintuitive in my opinion; a layer scoring the best should be capturing the structural information the best, shouldn't it? So that perturbing the best-scoring layer impact the model behavior the most. I believe there should be some consistency here, or more justification of why this is not the case. And if it is not the case, there needs to be some systematic way to determine which layer to intervene.

---

> > > ### Author Response · Authors · 2025-12-12
> > >
> > > We thank the reviewer for bringing this issue up. We acknowledge we did not perform an extensive study on the most effective intervention location, therefore we can only present situational evidence. In our observations, intervening on later layers, and thus letting the model propagate the manifold, made the response more robust, therefore lowering intervention success. Follow-up works geared towards pure intervention would clarify what the optimal setup is.
> > >
> > > However, even without optimizing for the most destructive intervention we still observe a drastic drop in performance. We believe this fact, along with the analysis we presented, to be sufficient evidence to validate our claim that feature manifolds are instrumental for reasoning.

---

### Author Response · Authors · 2025-12-10

As the decision period is drawing to a close, we kindly invite all reviewers to share any follow-up thoughts on our rebuttals. Let us know if you have any remaining questions or need anything clarified. We also hope our responses have addressed your concerns.

Thank you again for your time and engagement!

---

> ### Author Response · Authors · 2026-01-04
>
> We respectfully ask to be informed of the final acceptance decision if the review process is complete. Thank you again for your time and consideration.

---

### Decision · Action_Editor_BNL1 · 2026-01-05

**Recommendation:** Accept with minor revision

**Additional Comments:**

The paper can be accepted, but to meet the standard TMLR expectations on robustness, evidence, and practicality please implement  the following revisions :

!. Please downplay “automatic manifold discovery” claims (Title/Abstract/Intro/Experiments/Conclusion). Align language with what the method actually does: hypothesis-driven model selection over a user-specified set of candidate label-distance hypotheses $d_h$, using a linear map W. Remove “automatic” / “discover” phrasing. Restate findings consistent with what is actually guaranteed as “hypothesis-driven manifold identification”/“testing candidate geometric hypotheses”/“linear geometry probing aligned to hypothesized label distances”.
2. Add mentioning, at least in related works, of the nonlinear topology preserving dimensionality reduction baseline:  RTD AE (arXiv:2302.00136),  with its lighter version RTD-Lite AE (arXiv:2503.11910).
3. Strengthen manifold-selection reliability and identifiability analyses.
Directly address the paper’s own observation that stress values can be tightly clustered, making the “best” manifold hard to distinguish. Add robustness checks: bootstrap/subsampling CIs for Delta-stress, sensitivity to latent dimension m, data splits; and introduce an explicit “multiple plausible manifolds” decision rule when differences are not practically/statistically meaningful.

**Audience:**

Yes

**Audience Explanation:**

All three reviewers confirmed that the paper is interesting to at least some readers in TMLR's audience.

**Claims And Evidence:**

Yes

**Claims Explanation:**

The paper can be accepted, but to meet the standard TMLR expectations on robustness, evidence, and practicality please implement  the following revisions:

!. Please downplay “automatic manifold discovery” claims (Title/Abstract/Intro/Experiments/Conclusion). Align language with what the method actually does: hypothesis-driven model selection over a user-specified set of candidate label-distance hypotheses $d_h$, using a linear map W. Remove “automatic” / “discover” phrasing. Restate findings consistent with what is actually guaranteed as “hypothesis-driven manifold identification”/“testing candidate geometric hypotheses”/“linear geometry probing aligned to hypothesized label distances”.
2. Add mentioning, at least in related works, of the nonlinear topology preserving dimensionality reduction baseline:  RTD AE (arXiv:2302.00136),  with its lighter version RTD-Lite AE (arXiv:2503.11910).
3. Strengthen manifold-selection reliability and identifiability analyses.
Directly address the paper’s own observation that stress values can be tightly clustered, making the “best” manifold hard to distinguish. Add robustness checks: bootstrap/subsampling CIs for Delta-stress, sensitivity to latent dimension m, data splits; and introduce an explicit “multiple plausible manifolds” decision rule when differences are not practically/statistically meaningful.

---

> ### Author Response · Authors · 2026-02-04
>
> We sincerely thank the Action Editor and the Reviewers for their recognition of our work and their many insightful comments. We have provided an updated version of the manuscript implementing the latest suggestions of the Action Editor. In summary, we have:
> - Rephrased "Feature Manifold Discovery" as "Feature Manifold Analysis" and made more clear the scope of SMDS lies in evaluating specific geometric hypotheses;
> - Added references to RTD AE and RTD-Lite AE in the related works;
> - Established stronger statistical guarantees via bootstrapping, identifying a single statistically significant manifold for all datasets of the study (save for date_temperature whose case we discuss). The statistical analysis and its Confidence Intervals are discussed in the paper and in a new appendix;
> - Added a study on sensitivity to dimensionality $m$;
> - Expanded the discussion on how statistical significance can be used as a decision rule for best manifold. In cases where a single best manifold cannot be identified, we recommend developing new hypotheses and running further causal studies;
>
> We are at your disposal for any other clarification or suggestion to help improve the manuscript.